# Joint Identification and Sensing for Discrete Memoryless Channels

**DOI:** 10.3390/e27010012

**Published:** 2024-12-27

**Authors:** Wafa Labidi, Yaning Zhao, Christian Deppe, Holger Boche

**Affiliations:** 1Chair of Theoretical Information Technology, Technical University of Munich, 80333 Munich, Germany; wafa.labidi@tum.de (W.L.); boche@tum.de (H.B.); 2Institute for Communications Technology, Technische Universität Braunschweig, 38106 Brunswick, Germany; christian.deppe@tu-bs.de

**Keywords:** joint identification and sensing, message identification, information theory

## Abstract

In the identification (ID) scheme proposed by Ahlswede and Dueck, the receiver’s goal is simply to verify whether a specific message of interest was sent. Unlike Shannon’s transmission codes, which aim for message decoding, ID codes for a discrete memoryless channel (DMC) are far more efficient; their size grows doubly exponentially with the blocklength when randomized encoding is used. This indicates that when the receiver’s objective does not require decoding, the ID paradigm is significantly more efficient than traditional Shannon transmission in terms of both energy consumption and hardware complexity. Further benefits of ID schemes can be realized by leveraging additional resources such as feedback. In this work, we address the problem of joint ID and channel state estimation over a DMC with independent and identically distributed (i.i.d.) state sequences. State estimation functions as the sensing mechanism of the model. Specifically, the sender transmits an ID message over the DMC while simultaneously estimating the channel state through strictly causal observations of the channel output. Importantly, the random channel state is unknown to both the sender and the receiver. For this system model, we present a complete characterization of the ID capacity–distortion function.

## 1. Introduction

The identification (ID) scheme suggested by Ahlswede and Dueck [1] in 1989 is conceptually different from the classical message transmission scheme proposed by Shannon [2]. In classical message transmission, the encoder transmits a message over a noisy channel; at the receiver side, the aim of the decoder is to output an estimation of this message based on the channel observation. In the ID paradigm, however, the encoder sends an ID message (also called the identity) over a noisy channel, and the decoder aims to check whether or not a specific ID message of special interest to the receiver has been sent. Obviously, the sender has no prior knowledge of the specific ID message that the receiver is interested in. Ahlswede and Dueck demonstrated that in the theory of ID [1], if randomized encoding is used, then the size of ID codes for discrete memoryless channels (DMCs) grows doubly exponentially fast with the blocklength. If only deterministic encoding is allowed, then the number of identities that can be identified over a DMC scales exponentially with the blocklength. Nevertheless, the rate is still more significant than the transmission rate in the exponential scale, as shown in [3,4].

New applications in modern communications demand high reliability and latency requirements, including machine-to-machine and human-to-machine systems, digital watermarking [5,6,7], industry 4.0 [8], and 6G communication systems [9,10]. The aforementioned requirements are crucial for achieving trustworthiness [11]. For this purpose, the necessary latency resilience and data security requirements must be embedded in the physical domain. In this situation, the classical Shannon message transmission is limited and an ID scheme can achieve a better scaling behavior in terms of necessary energy and needed hardware components. It has been proved that information-theoretic security can be integrated into the ID scheme without paying an extra price for secrecy [12,13]. Further gains within the ID paradigm can be achieved by taking advantage of additional resources such as quantum entanglement, common randomness (CR), and feedback. In contrast to the classical Shannon message transmission, feedback can increase the ID capacity of a DMC [14]. Furthermore, it has been shown in [15] that the ID capacity of Gaussian channels with noiseless feedback is infinite. This holds to both rate definitions $\frac{1}{n}\log M$ (as defined by Shannon for classical transmission) and $\frac{1}{n}\log\log M$ (as defined by Ahlswede and Dueck for ID over DMCs). Interestingly, the authors of [15] showed that the ID capacity with noiseless feedback remains infinite regardless of the scaling used for the rate, e.g., double exponential, triple exponential, etc. In addition, the resource CR allows for a considerable increase in the ID capacity of channels [6,16,17]. The aforementioned communication scenarios emphasize that the ID capacity has completely different behavior than Shannon’s capacity.

A key technology within 6G communication systems is the joint design of radio communication and sensor technology [11]. This enables the realization of revolutionary end-user applications [18]. Joint communication and radar/radio sensing (JCAS) means that sensing and communication are jointly designed based on sharing the same bandwidth. Sensing and communication systems are usually designed separately, meaning that resources are dedicated to either sensing or data communications. The joint sensing and communication approach is a solution that can overcome the limitations of separation-based approaches. Recent works [19,20,21,22] have explored JCAS and showed that this approach can improve spectrum efficiency while minimizing hardware costs. For instance, the fundamental limits of joint sensing and communication for a point-to-point channel were studied in [23]. In this case, the transmitter wishes to simultaneously send a message to the receiver and sense its channel state through a strictly causal feedback link. Motivated by the drastic effects of feedback on the ID capacity [15], this work investigates joint ID and sensing. To the best of our knowledge, the problem of joint ID and sensing has not been treated in the literature yet. We study the problem of joint ID and channel state estimation over a DMC with i.i.d. state sequences. The sender simultaneously sends an ID message over the DMC with a random state and estimates the channel state via a strictly causal channel output. The random channel state is available to neither the sender nor the receiver. We consider the ID capacity–distortion tradeoff as a performance metric. This metric is analogous to the one studied in [24], and is defined as the supremum of all achievable ID rates such that some distortion constraint on state sensing is fulfilled. This model was motivated by the problem of adaptive and sequential optimization of the beamforming vectors during the initial access phase of communication [25]. We establish a lower bound on the ID capacity–distortion tradeoff. In addition, we show that in our communication setup, sensing can be viewed as an additional resource that increases the ID capacity.

Outline:The remainder of this paper is organized as follows. Section 2 introduces the system models, reviews key definitions related to identification (ID), and presents the main results, including a complete characterization of the ID capacity–distortion function. Section 3 provides detailed proofs of these main results. In Section 4, we explore an alternative more flexible distortion constraint, namely, the average distortion, and establish a lower bound on the corresponding ID capacity–distortion function. Finally, Section 5 concludes the paper with a discussion of the results and potential directions for future research.

Notation: The distribution of an RV *X* is denoted by $P_{X}$; for a finite set X, we denote the set of probability distributions on X by P(X) and the cardinality of X by |X|. If *X* is a RV with distribution $P_{X}$, we denote the Shannon entropy of *X* by $H(P_{X})$, the expectation of *X* by E(X), and the variance of *X* by Var[X]. If *X* and *Y* are two RVs with probability distributions $P_{X}$ and $P_{Y}$, then the mutual information between *X* and *Y* is denoted by I(X;Y). Finally, $X^{c}$ denotes the complement of X, X−Y denotes the difference set, and all logarithms and information quantities are taken to base 2.

## 2. System Models and Main Results

Consider a discrete memoryless channel with random state $\left(X\times S,W_{S}\left(y|x,s\right),Y\right)$ consisting of a finite input alphabet X, finite output alphabet Y, finite state set S, and pmf W(y|x,s) on Y. The channel is memoryless, i.e., if the input sequence $x^{n}\in X^{n}$ is sent and the sequence state is $s^{n}\in S^{n}$, then the probability of a sequence $y^{n}\in Y^{n}$ being received is provided by
(1)$$W_{S}^{n}\left(y^{n}|x^{n},s^{n}\right)=\prod_{i=1}^{n}W_{S}\left(y_{i}|x_{i},s_{i}\right).$$
The state sequence $\left(S_{1},S_{2},\ldots ,S_{n}\right)$ is i.i.d. according to the distribution $P_{S}$. We assume that the input $X_{i}$ and state $S_{i}$ are statistically independent for all i∈{1,2,…,n}. In our settup depicted in Figure 1, we assume that the channel state is known to neither the sender nor the receiver.

In the sequelae, we distinguish three scenarios:Randomized ID over the state-dependent channel $W_{S}$, as depicted in Figure 1,Deterministic or randomized ID over the state-dependent channel $W_{S}$ in the presence of noiseless feedback between the sender and the receiver, as depicted in Figure 2,Joint deterministic or randomized ID and sensing, in which the sender wishes to simultaneously send an identity to the receiver and sense the channel state sequence based on the output of the noiseless feedback link, as depicted in Figure 3.

First, we define randomized ID codes for the state-dependent channel defined above.

**Definition 1.** 
*An $\left(n,N,\lambda_{1},\lambda_{2}\right)$ randomized ID code with $\lambda_{1}+\lambda_{2}<1$ for channel $W_{S}$ is a family of pairs $\left\{{\left(Q\left(\cdot |i\right),D_{i}\left(s^{n}\right)\right)}_{s^{n}\in S^{n}},\;i=1,\ldots ,N\right\}$ with*

(2)
$$Q\left(\cdot |i\right)\in P\left(X^{n}\right),\;D_{i}\left(s^{n}\right)\in Y^{n},\;\forall s^{n}\in S^{n},\;\forall i=1,\ldots ,N,$$

*such that the errors of the first kind and the second kind are bounded as follows:*

(3)
$$\begin{array}{cc} \sum_{s^{n}\in S^{n}}P_{S}^{n}\left(s^{n}\right)\sum_{x^{n}\in X^{n}}Q\left(x^{n}|i\right)W_{S}^{n}\left(D_{i}{\left(s^{n}\right)}^{c}|x^{n},s^{n}\right) & \leq \lambda_{1},\forall i, \end{array}$$


(4)
$$\begin{array}{cc} \sum_{s^{n}\in S^{n}}P_{S}^{n}\left(s^{n}\right)\sum_{x^{n}\in X^{n}}Q\left(x^{n}|i\right)W_{S}^{n}\left(D_{j}\left(s^{n}\right)|x^{n},s^{n}\right) & \leq \lambda_{2},\forall i\neq j. \end{array}$$



In the following, we define the achievable ID rate and ID capacity for our system model.

**Definition 2.** 
*1.* 
*The rate R of a randomized $\left(n,N,\lambda_{1},\lambda_{2}\right)$ ID code for the channel $W_{S}$ is $R=\frac{\log\log(N)}{n}$ bits.*
*2.* 
*The ID rate R for $W_{S}$ is said to be achievable if for $\lambda \in (0,\frac{1}{2})$ there exists an $n_{0}\left(\lambda \right)$ such that for all $n\geq n_{0}\left(\lambda \right)$ there exists an $\left(n,2^{2^{nR}},\lambda ,\lambda \right)$ randomized ID code for $W_{S}$.*
*3.* 
*The randomized ID capacity $C_{ID}\left(W_{S}\right)$ of the channel $W_{S}$ is the supremum of all achievable rates.*



The following theorem characterizes the randomized ID capacity of the state-dependent channel $W_{S}$ when the state information is known to neither the sender nor the receiver.

**Theorem 1.** 
*The randomized ID capacity of the channel $W_{S}$ is provided by*

(5)
$$C_{ID}\left(W_{S}\right)=C\left(W_{S}\right)=\max_{P_{X}\in P\left(X\right)}I\left(X;Y\right),$$

*where $C(W_{S})$ denotes the Shannon transmission capacity of $W_{S}$.*


**Proof.** The proof of Theorem 1 follows from Theorem 6.6.4 of [26] and Equation (7.2) of [27]. Because the channel $W_{S}$ satisfies the strong converse property [26], the randomized ID capacity of $W_{S}$ coincides with its Shannon transmission capacity determined in [27]. □

Now, we consider the second scenario depicted in Figure 2. Let further ${\bar{Y}}^{n}=\left({\bar{Y}}_{1},\ldots ,{\bar{Y}}_{n}\right)\in Y^{n}$ denote the output of the noiseless backward (feedback) channel:(6)$${\bar{Y}}_{t}=Y_{t},\;\forall t\in \left\{1,\ldots ,n\right\}.$$
In the following, we define a deterministic and randomized ID feedback code for the state-dependent channel $W_{S}$.

**Definition 3.** 
*An $\left(n,N,\lambda_{1},\lambda_{2}\right)$ deterministic ID feedback code $\left\{{\left(f_{i},D_{i}\left(s^{n}\right)\right)}_{s^{n}\in S^{n}},\;i=1,\ldots ,N\right\}$ with $\lambda_{1}+\lambda_{2}<1$ for the channel $W_{S}$ is characterized as follows. The sender wants to send an ID message i∈N:={1,…,N} that is encoded by the vector-valued function*

(7)
$$\begin{array}{c} f_{i}=\left[f_{i}^{1},f_{i}^{2}\ldots ,f_{i}^{n}\right], \end{array}$$

*where $f_{i}^{1}\in X$ and $f_{i}^{t}:Y^{t-1}⟶X$ for t∈{2,…,n}. At t=1, the sender sends $f_{i}^{1}$. At t∈{2,…,n}, the sender sends $f_{i}^{t}\left(Y_{1},\ldots ,Y^{t-1}\right)$. The decoding sets $D_{i}\left(s^{n}\right)\subset Y^{n},\;\forall i\in \left\{1,\ldots ,N\right\},\mathrm{and}\;\forall s^{n}\in S^{n}$ should satisfy the following inequalities:*

(8)
$$\begin{array}{cc} \sum_{s^{n}\in S^{n}}P_{S}^{n}\left(s^{n}\right)W_{S}^{n}\left(D_{i}{\left(s^{n}\right)}^{c}|f_{i},s^{n}\right) & \leq \lambda_{1}\;\forall i, \end{array}$$


(9)
$$\begin{array}{cc} \sum_{s^{n}\in S^{n}}P_{S}^{n}\left(s^{n}\right)W_{S}^{n}\left(D_{j}\left(s^{n}\right)|f_{i},s^{n}\right) & \leq \lambda_{2}\;\forall i\neq j. \end{array}$$



**Definition 4.** 
*An $\left(n,N,\lambda_{1},\lambda_{2}\right)$ randomized ID feedback code*

$$\left\{{\left(Q_{F}\left(\cdot |i\right),D_{i}\left(s^{n}\right)\right)}_{s^{n}\in S^{n}},\;i=1,\ldots ,N\right\}$$

*with $\lambda_{1}+\lambda_{2}<1$ for the channel $W_{S}$ is characterized as follows. The sender wants to send an ID message i∈N:={1,…,N} that is encoded by the probability distribution*

(10)
$$\begin{array}{c} Q_{F}\left(\cdot |i\right)\in P\left(F^{n}\right), \end{array}$$

*where $Q_{F}\left(\cdot |i\right)$ denotes the set of all n-length functions f as $F^{n}$. The decoding sets $D_{i}\left(s^{n}\right)\subset Y^{n},\;\forall i\in \left\{1,\ldots ,N\right\}$, $\mathrm{and}\;\forall s^{n}\in S^{n}$ should satisfy the following inequalities:*

(11)
$$\begin{array}{cc} \sum_{s^{n}\in S^{n}}P_{S}^{n}\left(s^{n}\right)\sum_{f\in F^{n}}Q_{F}\left(f|i\right)W_{S}^{n}\left(D_{i}{\left(s^{n}\right)}^{c}|f,s^{n}\right) & \leq \lambda_{1}\;\forall i, \end{array}$$


(12)
$$\begin{array}{cc} \sum_{s^{n}\in S^{n}}P_{S}^{n}\left(s^{n}\right)\sum_{f\in F^{n}}Q_{F}\left(f|i\right)W_{S}^{n}\left(D_{j}\left(s^{n}\right)|f,s^{n}\right) & \leq \lambda_{2}\;\forall i\neq j. \end{array}$$



**Definition 5.** 
*1.* 
*The rate R of a (deterministic/randomized) $\left(n,N,\lambda_{1},\lambda_{2}\right)$ ID feedback code for the channel $W_{S}$ is $R=\frac{\log\log(N)}{n}$ bits.*
*2.* 
*The (deterministic/randomized) ID feedback rate R for $W_{S}$ is said to be achievable if for $\lambda \in (0,\frac{1}{2})$ there exists an $n_{0}\left(\lambda \right)$ such that for all $n\geq n_{0}\left(\lambda \right)$ there exists a (deterministic/randomized) $\left(n,2^{2^{nR}},\lambda ,\lambda \right)$ ID feedback code for $W_{S}$.*
*3.* 
*The (deterministic/randomized) ID feedback capacity $C_{IDf}^{d}\left(W_{S}\right)$/$C_{IDf}^{r}\left(W_{S}\right)$ of the channel $W_{S}$ is the supremum of all achievable rates.*



It has been demonstrated in [26] that noise increases the ID capacity of the DMC in the case of feedback. Intuitively, noise is considered a source of randomness, i.e., a random experiment for which outcome is provided to the sender and receiver via the feedback channel. Thus, adding a perfect feedback link enables the realization of a correlated random experiment between the sender and the receiver. The size of this random experiment can be used to compute the growth of the ID rate. This result has been further emphasized in [15,28], where it has been shown that the ID capacity of the Gaussian channel with noiseless feedback is infinite. This is because the authors of [15,28] provided a coding scheme that generates infinite common randomness between the sender and the receiver. Here, we want to investigate the effect of feedback on the ID capacity of the system model depicted in Figure 2. Theorem 2 characterizes the ID feedback capacity of the state-dependent channel $W_{S}$ with noiseless feedback. The proof of Theorem 2 is provided in Section 3.

**Theorem 2.** 
*If $C(W_{S})>0$, then the deterministic ID feedback capacity of $W_{S}$ is provided by*

(13)
$$C_{IDf}^{d}\left(W_{S}\right)=\max_{x\in X}H\left(E\left[W_{S}\left(\cdot |x,S\right)\right]\right).$$



**Theorem 3.** 
*If $C(W_{S})>0$, then the randomized ID feedback capacity of $W_{S}$ is provided by*

(14)
$$C_{IDf}^{r}\left(W_{S}\right)=\max_{P\in P\left(X\right)}H\left(\sum_{x\in X}P\left(x\right)E\left[W_{S}\left(\cdot |x,S\right)\right]\right).$$



**Remark 1.** 
*It can be shown that the same ID feedback capacity formula holds if the channel state is known to either the sender or the receiver. This is because we achieve the same amount of common randomness as in the scenario depicted in Figure 2. Intuitively, the channel state is an additional source of randomness that we can take advantage of.*


Now, we consider the third scenario depicted in Figure 3, where we want to jointly identify and sense the channel state. The sender comprises an encoder that sends a symbol $x_{t}=f_{i}^{t}\left(y^{t-1}\right)$ for each identity i∈{1,…,N} and delayed feedback output $y^{t-1}\in Y^{t-1}$ along with a state estimator that outputs an estimation sequence ${\hat{s}}^{n}\in S^{n}$ based on the feedback output and input sequence. We define the per-symbol distortion as follows:(15)$$d_{t}=E\left[d(S_{t},{\hat{S}}_{t})\right],$$
where d:S×S→[0,+∞) is a distortion function and the expectation is over the joint distribution of $\left(S_{t},{\hat{S}}_{t}\right)$ conditioned by the ID message i∈{1,…,N}.

**Definition 6.** 
*1.* 
*An ID rate–distortion pair (R,D) for $W_{S}$ is said to be achievable if for every $\lambda \in (0,\frac{1}{2})$ there exists an $n_{0}\left(\lambda \right)$ such that for all $n\geq n_{0}\left(\lambda \right)$ there exists an $\left(n,2^{2nR},\lambda ,\lambda \right)$ (deterministic/randomized) ID code for $W_{S}$ and if $d_{t}\leq D$ for all t=1,⋯,n.*
*2.* 
*The deterministic ID capacity–distortion function $C_{ID}^{d}\left(D\right)$ is defined as the supremum of R such that (R,D) is achievable.*



Without loss of generality, we choose the following deterministic estimation function $h^{★}$:(16)$$\hat{s}=h^{★}\left(x,y\right)=\min_{h:X\times Y\to S}E\left[d(S,h(X,Y))|X=x,Y=y\right],$$
where h:X×Y→S is an estimator that maps a channel input–feedback output pair to a channel state. If there exist several functions $h^{★}\left(\cdot ,\cdot \right)$, we choose one randomly. We define the minimal distortion function for each input symbol x∈X as in [29]:(17)$$d^{★}\left(x\right)=E_{SY}\left[d\left(S,h^{*}\left(X,Y\right)\right)|X=x\right]$$
and the minimal distortion function for each input distribution $P\in P\left(X\right)$ as
(18)$$d^{★}\left(P\right)=\sum_{x\in X}d^{★}\left(x\right).$$
In the following, we establish the ID capacity–distortion function defined above.

**Theorem 4.** 
*The deterministic ID capacity–distortion function of the state-dependent channel $W_{S}$ depicted in Figure 3 is provided by*

(19)
$$C_{ID}^{d}\left(D\right)=\max_{x\in X_{D}}H\left(E[W_{S}\left(\cdot |x,S\right)]\right),$$

*where the set $X_{D}$ is provided by*

(20)
$$X_{D}=\left\{x\in X,\;d^{★}\left(x\right)\leq D\right\}.$$



We now turn our attention to a randomized encoder. In the following, we derive the ID capacity–distortion function of the state-dependent channel $W_{S}$ under the assumption of randomized encoding.

**Theorem 5.** 
*The randomized ID capacity–distortion function of the state-dependent channel $W_{S}$ is provided by*

(21)
$$C_{ID}^{r}\left(D\right)=\max_{P\in P_{D}}H\left(\sum_{x\in X}P\left(x\right)E\left[W_{S}\left(\cdot |x,S\right)\right]\right),$$

*where the set $P_{D}$ is provided by*

(22)
$$P_{D}=\left\{P\in P\left(X\right),\;d^{★}\left(P\right)\leq D\right\}.$$



**Remark 2.** 
*Randomized encoding achieves higher rates than deterministic encoding. This is because we are combining two sources of randomness: local randomness used for the encoding, and shared randomness generated via the noiseless feedback link. The result is analogous to randomized ID over DMCs in the presence of noiseless feedback, as studied in [14].*


## 3. Proof of the Main Results

In this section, we provide the proofs of Theorems 2–5.

### 3.1. Direct Proof of Theorem 2

**Proof.** We consider an average channel $W_{S,\mathrm{avg}}$ provided by
(23)$$W_{S,\mathrm{avg}}\left(y|x\right)=\sum_{s\in S}P_{S}\left(s\right)W_{S}\left(y|x,s\right),\;\forall x\in X,\;\forall y\in Y.$$
The DMC $W_{S,\mathrm{avg}}$ is obtained by averaging the DMCs $W_{S}$ over the state. Now, it suffices to show that $R=\max_{x\in X}H\left(E[W_{S}\left(\cdot |x,S\right)]\right)$ is an achievable ID feedback rate for the average channel $W_{S}^{a}$. The deterministic ID feedback capacity of the average channel $C_{IDf}^{d}\left(W_{S,\mathrm{avg}}\right)$ can be determined by applying Theorem 1 of [28] on $W_{S,\mathrm{avg}}$. If the transmission capacity $C(W_{S,\mathrm{avg}})$ of $W_{S,\mathrm{avg}}$ is positive, we have
(24)$$\begin{array}{cc} C_{IDf}^{d}\left(W_{S,\mathrm{avg}}\right) & \leq \max_{x\in X}H\left(W_{S,\mathrm{avg}}\left(\cdot |x\right)\right) \end{array}$$
(25)$$\begin{array}{cc} \; & \;\;\;\;\;=\max_{x\in X}H\left(E[W_{S}\left(\cdot |x,S\right)]\right). \end{array}$$
This completes the direct proof of Theorem 2. □

### 3.2. Converse Proof of Theorem 2

**Proof.** For the converse proof, we use the techniques of [14] for deterministic ID over DMCs with noiseless feedback. We first extend Lemma 3 of [14] (image size for a deterministic feedback strategy) to the deterministic ID feedback code for $W_{S}$ described in Definition 3. □

**Lemma 1.** 
*For any feedback strategy $f=[f^{1},f^{2}\ldots ,f^{n}]$ and any μ∈(0,1), we have*

(26)
$$\min_{E_{1}\subset Y^{n}:E_{S^{n}}\left[W_{S}^{n}\left(E_{1}|f,S^{n}\right)\right]\geq 1-\mu }\left|E_{1}\right|\leq K_{1},$$

*where $K_{1}$ is provided by*

(27)
$$K_{1}=2^{n\max_{x\in X}H\left(E\left[W_{S}\left(\cdot |x\right)\right]\right)+\alpha \sqrt{n}}=2^{nH\left(E\left[W_{S}\left(\cdot |x^{★}\right)\right]\right)+\alpha \sqrt{n}},$$

*where*

(28)
$$\alpha =\sqrt{\frac{\beta }{\mu }},\;\beta =\max(\log^{2}\left(3\right),\log^{2}\left(|Y|\right).$$



**Proof.** We use a similar idea as for the proof of Lemma 3 of [14]. Let $E_{1}^{★}\subset Y^{n}$ be defined as follows:
(29)$$E_{1}^{★}=\left\{y^{n}\in Y^{n},\;-\log E_{S^{n}}\left[W_{S}^{n}\left(y^{n}|f,S^{n}\right)\right]\leq \log K_{1}\right\}.$$
It then follows from the definition of $E_{1}^{★}$ that $|E^{★}|\leq K_{1}$. It remains to show that
$$E_{S^{n}}\left[W_{S}^{n}\left(y^{n}|f,S^{n}\right)\right]\geq 1-\mu .$$
For t=1,2,…,n and a fixed feedback strategy $f=[f^{1},\ldots ,f^{t}]\in X$, we have
(30)$$\begin{array}{cc} \mathrm{Pr}\{Y^{t}=y^{t}\} & =\sum_{s^{n}\in S^{n}}P_{S}^{n}\left(s^{n}\right)W^{n}\left(y^{n}|f,s^{n}\right) \end{array}$$
(31)$$\begin{array}{cc} \; & \;\;\;\;\;\;\;=E_{S^{n}}\left[W_{S}^{n}\left(y^{n}|f,S^{n}\right)\right],\;\forall y^{t}\in Y^{t}. \end{array}$$
Let the RV $Z_{t}$ be defined as follows:
(32)$$Z_{t}=-\log E_{S_{t}}\left[W_{S}\left(Y_{t}\right|f^{t}\left(Y^{\left(t-1\right)},S_{t}\right)\right].$$
We have
(33)$$\begin{array}{cc} \; & E_{S^{n}}\left[W_{S}^{n}\left(E_{1}^{★}|f,S^{n}\right)\right] \end{array}$$
(34)$$\begin{array}{cc} \; & =\mathrm{Pr}\left\{-\log E_{S^{n}}\left[W_{S}^{n}\left(y^{n}|f,S^{n}\right)\right]\leq \log K_{1}\right\} \end{array}$$
(35)$$\begin{array}{cc} \; & =\mathrm{Pr}\left\{-\log\left(\sum_{s^{n}\in S^{n}}P_{S}^{n}\left(s^{n}\right)\prod_{t=1}^{n}W_{S}\left(y_{t}|f^{t},s_{t}\right)\right)\leq \log K_{1}\right\} \end{array}$$
(36)$$\begin{array}{cc} \; & =\mathrm{Pr}\left\{-\log\left(\sum_{s_{1}\in S}\sum_{s_{2}\in S}\cdots \sum_{s_{n}\in S}\prod_{t=1}^{n}P_{S}\left(s_{t}\right)W_{S}\left(y_{t}|f^{t},s_{t}\right)\right)\leq \log K_{1}\right\} \end{array}$$
(37)$$\begin{array}{cc} \; & =\mathrm{Pr}\{-\log(\sum_{s_{1}\in S}P_{S}\left(s_{1}\right)W_{S}\left(y_{1}|f^{1},s_{1}\right) \end{array}$$
(38)$$\begin{array}{cc} \; & \;\cdot \sum_{s_{2}\in S}P_{S}\left(s_{2}\right)W_{S}\left(y_{2}|f^{2},s_{2}\right)\cdots \sum_{s_{n}\in S}P_{S}\left(s_{n}\right)W_{S}\left(y_{n}|f^{n},s_{n}\right))\leq \log K_{1}\} \end{array}$$
(39)$$\begin{array}{cc} \; & =\mathrm{Pr}\left\{\sum_{t=1}^{n}-\log E_{S_{t}}\left[W_{S}\left(Y_{t}\right|f^{t}\left(Y^{\left(t-1\right)},S_{t}\right)\right]\leq \log K_{1}\right\} \end{array}$$
(40)$$\begin{array}{cc} \; & =\mathrm{Pr}\left\{\sum_{t=1}^{n}Z_{t}\leq \log K_{1}\right\}. \end{array}$$
Now, we want to establish a lower bound on $E_{S^{n}}\left[W_{S}^{n}\left(E^{★}|f,S^{n}\right)\right]$. It suffices to find a lower bound on the expression in (40). Let the RV $U_{t}$ be defined as follows:
(41)$$U_{t}=Z_{t}-E\left[Z_{t}|Y^{t-1}\right],\;t\in \left\{1,2,\ldots ,n\right\}.$$
It can be shown that
(42)$$\begin{array}{cc} E[U_{t}|Y^{t-1}] & =E_{Y}\left[Z_{t}-E_{Y}\left[Z_{t}|Y^{t-1}\right]|Y^{t-1}\right] \end{array}$$
(43)$$\begin{array}{cc} \; & =E_{Y}\left[Z_{t}|Y^{t-1}\right]-E_{Y}\left[Z_{t}|Y^{t-1}\right] \end{array}$$
(44)$$\begin{array}{cc} \; & =0. \end{array}$$
Furthermore, we have
(45)$$\begin{array}{cc} E[U_{t}] & =E_{Y}\left[Z_{t}-E_{Y}\left[Z_{t}|Y^{t-1}\right]\right] \end{array}$$
(46)$$\begin{array}{cc} \; & =E_{Y}\left[Z_{t}\right]-E_{Y}\left[E_{Y}\left[Z_{t}|Y^{t-1}\right]\right] \end{array}$$
(47)$$\begin{array}{cc} \; & =E_{Y}\left[Z_{t}\right]-E_{Y}\left[Z_{t}\right] \end{array}$$
(48)$$\begin{array}{cc} \; & =0. \end{array}$$
It can be shown that for $y^{t-1}\in Y^{t-1}$,
(49)$$\begin{array}{cc} E[Z_{t}|y^{t-1}]\; & =\sum_{y_{t}\in Y}(-E_{S_{t}}\left[W_{S}\left(y_{t}|f^{t}\left(y^{t-1}\right),S_{t}\right)\right] \end{array}$$
(50)$$\begin{array}{cc} \; & \;\log E_{S_{t}}\left[W_{S}\left(y_{t}|f^{t}\left(y^{t-1}\right),S_{t}\right)\right]) \end{array}$$
(51)$$\begin{array}{cc} \; & \leq \max_{x\in X}H\left(E_{S}\left[W_{S}\left(\cdot |x,S\right)\right]\right) \end{array}$$
(52)$$\begin{array}{cc} \; & =H\left(E_{S}\left[W_{S}\left(\cdot |x^{★},S\right)\right]\right). \end{array}$$
It follows from the definition of the RV $U_{t}$ in (41) that
(53)$$\begin{array}{cc} E_{S^{n}}\left[W_{S}^{n}\left(E^{★}|f,S^{n}\right)\right]\; & =\mathrm{Pr}\left\{\sum_{t=1}^{n}\left(U_{t}+E\left[Z_{t}|Y^{t-1}\right]\right)\leq \log K_{1}\right\} \end{array}$$
(54)$$\begin{array}{cc} \; & =\mathrm{Pr}\left\{\sum_{t=1}^{n}U_{t}\leq \log K_{1}-\sum_{t=1}^{n}E\left[Z_{t}|Y^{t-1}\right]\right\} \end{array}$$
(55)$$\begin{array}{cc} \; & \overset{\left(a\right)}{=}\mathrm{Pr}\{\sum_{t=1}^{n}U_{t}\leq nH\left(E_{S}\left[W_{S}\left(\cdot |x^{★},S\right)\right]\right) \end{array}$$
(56)$$\begin{array}{cc} \; & \;+\alpha \sqrt{n}-\sum_{t=1}^{n}E\left[Z_{t}|Y^{t-1}\right]\} \end{array}$$
(57)$$\begin{array}{cc} \; & \overset{\left(b\right)}{\geq }\mathrm{Pr}\left\{\sum_{t=1}^{n}U_{t}\leq \alpha \sqrt{n}\right\}, \end{array}$$
where (a) follows from the definition of $K_{1}$ in (27) and (b) follows from (52).It can be verified that
(58)$$\mathrm{Var}[U_{t}]\leq \beta ,\;\forall t=1,2,\ldots ,n.$$
Therefore, we can apply Chebyshev’s inequality and obtain
(59)$$\begin{array}{cc} E_{S^{n}}\left[W_{S}^{n}\left(E^{★}|f,S^{n}\right)\right] & \geq \mathrm{Pr}\left\{\sum_{t=1}^{n}U_{t}\leq \alpha \sqrt{n}\right\} \end{array}$$
(60)$$\begin{array}{cc} \; & \overset{\left(a\right)}{\geq }1-\mu , \end{array}$$
where (a) follows the definition of β in (28). This completes the proof of Lemma 1. □

We establish an upper bound on the deterministic ID feedback rate for the channel model $W_{S}$ using Lemma 1. Let $\left\{{\left(f_{i},D_{i}\left(s^{n}\right)\right)}_{s^{n}\in S^{n}},\;i=1,\ldots ,N\right\}$ be an (n,N,λ,λ) deterministic ID feedback code for channel $W_{S}$ with $\lambda \in (0,\frac{1}{2})$, and let μ∈(0,1) be chosen such that
(61)$$1-\mu -\lambda <\frac{1}{2}.$$
For each feedback strategy $f_{i}$, we define the set $E_{i}$ that satisfies (26). For i∈{1,2,…,N}, we have
(62)$$\begin{array}{cc} E_{S^{n}}\left[W_{S}^{n}\left(D_{i}\left(s^{n}\right)\cap E_{i}|f_{i},S^{n}\right)\right] & =1-E_{S^{n}}\left[W_{S}^{n}\left({\left(D_{i}\left(s^{n}\right)\cap E_{i}\right)}^{c}|f_{i},S^{n}\right)\right] \end{array}$$
(63)$$\begin{array}{cc} \; & =1-E_{S^{n}}\left[W_{S}^{n}\left({\left(D_{i}\left(s^{n}\right)\right)}^{c}\cup E_{i}^{c}|f_{i},S^{n}\right)\right] \end{array}$$
(64)$$\begin{array}{cc} \; & \overset{\left(a\right)}{\geq }1-E_{S^{n}}\left[W_{S}^{n}\left({\left(D_{i}\left(s^{n}\right)\right)}^{c}|f_{i},S^{n}\right)\right]-E_{S^{n}}\left[W_{S}^{n}\left(E_{i}^{c}|f_{i},S^{n}\right)\right] \end{array}$$
(65)$$\begin{array}{cc} \; & \overset{\left(b\right)}{\geq }1-\lambda -\mu \end{array}$$
(66)$$\begin{array}{cc} \; & \overset{\left(c\right)}{>}\frac{1}{2}, \end{array}$$
where (a) follows from the union bound, (b) follows from the definition of the (n,N,λ,λ) ID feedback code and from (26), and (c) follows from (61). Similarly, from the definition of the ID feedback code
$$\left\{{\left(f_{i},D_{i}\left(s^{n}\right)\right)}_{s^{n}\in S^{n}},\;i=1,\ldots ,N\right\},$$
from (26) and (61), for i∈{1,2,…,N} and i≠j it follows that
(67)$$E_{S^{n}}\left[W_{S}^{n}\left(D_{j}\left(s^{n}\right)\cap E_{j}|f_{i},S^{n}\right)\right]<\frac{1}{2}.$$
As the error of the second kind λ is smaller than $\frac{1}{2}$, all the sets $D_{i}\left(s^{n}\right)\cap E_{i}$ are distinct. Therefore, any (n,N,λ,λ) deterministic ID feedback code $\left\{{\left(f_{i},D_{i}\left(s^{n}\right)\right)}_{s^{n}\in S^{n}},\;i=1,\ldots ,N\right\}$ for channel $W_{S}$ with $\lambda \in (0,\frac{1}{2})$ has an associated $\left(n,N,\lambda^{\prime },\lambda^{\prime }\right)$ deterministic ID feedback code $\left\{{\left(f_{i},D_{i}\left(s^{n}\right)\right)}_{s^{n}\in S^{n}}\cap E_{i},\;i=1,\ldots ,N\right\}$, where $\lambda^{\prime }\in \left(0,\frac{1}{2}\right)$ and the set $E_{i}$ satisfies (26) ∀i=1,…,N. Thus, per Lemma 1, the cardinality *N* of the deterministic ID feedback code is upper-bounded as follows:
(68)N≤∑k=0K1|Y|nk≤|Y|nK1
(69)$$\begin{array}{cc} \; & =2^{n\log|Y|2^{nH\left(E[W_{S}\left(\cdot |x^{★},S\right)]\right)+\alpha \sqrt{n}}}. \end{array}$$
This completes the converse proof of Theorem 2.

### 3.3. Direct Proof of Theorem 3

**Proof.** Similarly, we can consider the average channel $W_{S,avg}$ defined in (23). It is sufficient to show that $R=\max_{P\in P(X)}H\left(\sum_{x\in X}P\left(x\right)W_{S,avg}\left(\cdot |x\right)\right)$ is an achievable randomized ID feedback rate for the average channel $W_{S,avg}$. The randomized ID feedback capacity of the average channel $C_{IDf}^{r}\left(W_{S,avg}\right)$ can be obtained by applying Theorem 2 of [28] on $W_{S,avg}$. If the transmission capacity $C(W_{S,avg})$ of $W_{S,avg}$ is positive, then we have
(70)$$\begin{array}{cc} C_{IDf}^{r}\left(W_{S,avg}\right)\; & \leq \max_{P\in P(X)}H\left(\sum_{x\in X}P\left(x\right)W_{S,avg}\left(\cdot |x\right)\right). \end{array}$$
This completes the direct proof of Theorem 3. □

### 3.4. Converse Proof of Theorem 3

**Proof.** We first extend Lemma 4 of [14] (image size for a randomized feedback strategy) to the randomized ID feedback code for $W_{S}$. □

**Lemma 2.** 
*For any randomized feedback strategy $Q_{F}\left(\cdot \right)$ over all n-length feedback encoding sets $F_{n}$ and any μ∈(0,1),*

(71)
$$\begin{array}{c} \min_{E_{2}\subset Y^{n}:\sum_{f\in F_{n}}Q_{F}\left(f\right)E_{S^{n}}\left[W_{S}^{n}\left(E_{2}|f,S^{n}\right)\right]\geq 1-\mu }\left|E_{2}\right|\leq K_{2}, \end{array}$$

*where $K_{2}$ is provided by*

(72)
$$\begin{array}{c} K_{2}=2^{n\max_{P\in P\left(X\right)}H\left(\sum_{x\in X}P\left(x\right)E\left[W_{S}\left(\cdot |x,S\right)\right]\right)+\alpha \sqrt{n}}=2^{nH\left(\sum_{x\in X}P^{★}\left(x\right)E\left[W_{S}\left(\cdot |x,S\right)\right]\right)+\alpha \sqrt{n}}, \end{array}$$

*where α=βμ, β=maxlog23,log2|Y|.*


**Proof.** We use a similar idea as for the proof of Lemma 4 in [14]. We define the set $E_{2}^{★}\subset Y^{n}$ as follows:
(73)$$\begin{array}{c} E_{2}^{★}=\left\{y^{n}\in Y^{n},-\log\sum_{f\in F_{n}}Q_{F}\left(f\right)E_{S^{n}}\left[W_{S}\left(y^{n}|f,S^{n}\right)\right]\leq \log K_{2}\right\}. \end{array}$$From the definition of $E_{2}^{★}$, we have
$$\left|E_{2}^{★}\right|\leq K_{2}.$$□

Then, it is sufficient to show that $\sum_{f\in F_{n}}Q_{F}\left(f\right)E_{S^{n}}\left[W_{S}^{n}\left(E_{2}|f,S^{n}\right)\right]\geq 1-\mu$. Let the RV $Z_{t}$ be defined as follows:
(74)$$Z_{t}=-\log\left(\sum_{f^{t}\in F^{t}}Q^{t}\left(f^{t}\right)E_{S_{t}}\left[W_{S}^{n}\left(Y_{t}|f^{t}\left(Y^{t-1}\right),S_{t}\right)\right]\right)$$
where $Q_{F}\left(f\right)=\prod_{t=1}^{n}Q^{t}\left(f^{t}\right)$ and $F^{t}$ is the set of all mapping $Y^{t-1}\mapsto X$. We have
(75)$$\begin{array}{cc} \; & \sum_{f\in F_{n}}Q_{F}\left(f\right)E_{S^{n}}\left[W_{S}^{n}\left(E_{2}|f,S^{n}\right)\right] \end{array}$$
(76)$$\begin{array}{cc} \; & =Pr\left\{-\log\left(\sum_{f\in F_{n}}Q_{F}\left(f\right)E_{S^{n}}\left[W_{S}\left(y^{n}|f,S^{n}\right)\right]\right)\leq \log K_{2}\right\} \end{array}$$
(77)$$\begin{array}{cc} \; & =Pr\left\{-\log\left(\sum_{f\in F_{n}}Q_{F}\left(f\right)\prod_{t=1}^{n}E_{S_{t}}\left[W_{S}\left(Y_{t}\right|f^{t}\left(Y^{t-1}\right),S_{t}\right]\right)\leq \log K_{2}\right\} \end{array}$$
(78)$$\begin{array}{cc} \; & =Pr\left\{-\log\left(\sum_{f\in F_{n}}\prod_{t=1}^{n}Q^{t}\left(f^{t}\right)\prod_{t=1}^{n}E_{S_{t}}\left[W_{S}\left(Y_{t}\right|f^{t}\left(Y^{t-1}\right),S_{t}\right]\right)\leq \log K_{2}\right\} \end{array}$$
(79)$$\begin{array}{cc} \; & =Pr\left\{-\log\left(\prod_{t=1}^{n}\sum_{f^{t}\in F^{t}}Q^{t}\left(f^{t}\right)E_{S_{t}}\left[W_{S}\left(Y_{t}\right|f^{t}\left(Y^{t-1}\right),S_{t}\right]\right)\leq \log K_{2}\right\} \end{array}$$
(80)$$\begin{array}{cc} \; & =Pr\left\{\sum_{t=1}^{n}Z_{t}\leq \log K_{2}\right\}. \end{array}$$
Now, for any $y^{t-1}\in Y^{t-1}$, we consider
(81)$$\begin{array}{cc} \; & E\left[Z_{t}|y^{t-1}\right] \end{array}$$
(82)$$\begin{array}{cc} \; & =\sum_{y_{t}\in Y}\left(-\sum_{f^{t}\in F^{t}}Q^{t}\left(f^{t}\right)E_{S_{t}}\left[W_{S}^{n}\left(y_{t}|f^{t}\left(y^{t-1}\right),S_{t}\right)\right]\log\left(\sum_{f^{t}\in F^{t}}Q^{t}\left(f^{t}\right)E_{S_{t}}\left[W_{S}^{n}\left(y_{t}|f^{t}\left(y^{t-1}\right),S_{t}\right)\right]\right)\right) \end{array}$$
(83)$$\begin{array}{cc} \; & \leq H\left(\sum_{x\in X}P^{★}\left(x\right)E\left[W_{S}^{n}\left(\cdot \right|x,S\right]\right). \end{array}$$
Similarly, for all $t\in \left\{1,2,\cdots ,n\right\}$, we define the RV $U_{t}=Z_{t}-E\left[Z_{t}|Y^{t-1}\right]$. It has been shown in (44) and (48) that $E\left[U_{t}|Y^{t-1}\right]=0$ and $E\left[U_{t}\right]=0$. Combining (80) and (83), we have
(84)$$\begin{array}{cc} \; & \sum_{f\in F_{n}}Q_{F}\left(f\right)E_{S^{n}}\left[W_{S}^{n}\left(E_{2}|f,S^{n}\right)\right] \end{array}$$
(85)$$\begin{array}{cc} \; & =Pr\left\{\sum_{t=1}^{n}\left(U_{t}+E\left[Z_{t}|Y^{t-1}\right]\right)\leq \log K_{2}\right\} \end{array}$$
(86)$$\begin{array}{cc} \; & =Pr\left\{\sum_{t=1}^{n}U_{t}\leq nH\left(\sum_{x\in X}P^{★}\left(x\right)E\left[W_{S}\left(\cdot |x,S\right)\right]\right)+\alpha \sqrt{n}-\sum_{t=1}^{n}E\left[Z_{t}|Y^{t-1}\right]\right\} \end{array}$$
(87)$$\begin{array}{cc} \; & \geq Pr\left\{\sum_{t=1}^{n}U_{t}\leq \alpha \sqrt{n}\right\}. \end{array}$$
Assuming $Var[U_{t}]\leq \beta$ for all t=1,2,⋯,n, we can apply Chebyshev’s inequality to obtain
(88)$$\begin{array}{cc} \sum_{f\in F_{n}}Q_{F}\left(f\right)E_{S^{n}}\left[W_{S}^{n}\left(E_{2}|f,S^{n}\right)\right]\; & \geq 1-\frac{\beta }{\alpha^{2}} \end{array}$$
(89)$$\begin{array}{cc} \; & =1-\mu . \end{array}$$
Replacing $K_{1}$ in the converse proof of Theorem 2 with the corresponding $K_{2}$ as outlined in Lemma 2 completes the converse proof of Theorem 3.

### 3.5. Direct Proof of Theorem 4

#### 3.5.1. Coding Scheme

**Proof.** To some extent, we use the same coding scheme elaborated in [14]. We choose the blocklength as $m=n+\lceil \sqrt{n}\rceil$. Let $x^{★}$ be some symbol in $X_{D}$. Regardless of which identity i∈{1,…,N} we want to identify, the sender first sends the sequence ${x^{★}}^{n}=\left(x^{★},x^{★},\ldots ,x^{★}\right)\in X_{D}^{n}$ over the state-dependent channel $W_{S}$. The received sequence $y^{n}\in Y^{n}$ becomes known to the sender (estimator and encoder) via the noiseless feedback link. The feedback provides the sender and receiver with knowledge of the outcome of the correlated random experiment $\left(Y^{n},E_{S^{n}}\left[W_{S}^{n}\left(\cdot |{x^{★}}^{n},S^{n}\right)\right]\right)$. □

#### 3.5.2. Common Randomness Generation

We want to generate uniform common randomness, as it is the most convenient form of common randomness [17]; therefore, we convert our correlated random experiment $\left(Y^{n},E_{S^{n}}\left[W_{S}^{n}\left(\cdot |{x^{★}}^{n},S^{n}\right)\right]\right)$ to a uniform one $\left(T^{n},E_{S^{n}}\left[W_{S}^{n}\left(\cdot |{x^{★}}^{n},S^{n}\right)\right]\right)$. For ϵ>0, the set $T^{n}$ is provided by
(90)$$T^{n}=\underset{V_{S}^{a}:∥V_{S}^{a}-W_{S}^{a}∥\leq \epsilon }{⋃}T_{V_{S}^{a}}^{n}\left({x^{★}}^{n}\right),$$
where $∥V_{S}^{a}-W_{S}^{a}∥$ is defined as follows.

**Definition 7.** 
*Let W be the set of stochastic matrices W:X→Y. Let W∈W such that*

(91)
$$P_{XY}\left(x,y\right)=P_{X}\left(x\right)W\left(y|x\right),\;\forall x\in X,\forall y\in Y.$$

*For $V,V^{\prime }\in W$, the distance $|V-V^{\prime }|$ is defined as*

(92)
$$∥V-V^{\prime }∥=\max_{x\in X,y\in Y}\left|V\left(y|x\right)-V^{\prime }\left(y|x\right)\right|.$$



Here, $V_{S}^{a}$ denotes the average channel defined by
(93)$$V_{S}^{a}=\sum_{s\in S}P_{S}\left(s\right)V_{S}\left(\cdot |\cdot ,s\right).$$
We introduce the following lemmas.

**Lemma 3** ([30])**.** 
*Let $\left(x^{n},y^{n}\right)$ be emitted by the DMS $P_{XY}\left(\cdot ,\cdot \right)=W\left(\cdot |\cdot \right)P_{X}\left(\cdot \right)$ and let V∈W such that |V−W|≤ϵ. Then, for every ϵ>0 there exist $\delta^{\prime }>0$ and $n_{0}\left(\epsilon \right)$ such that for $n\geq n_{0}\left(\epsilon \right)$ we have*
(94)$$\sum_{y^{n}\in T_{V}^{n}\left(x^{n}\right)}W^{n}\left(y^{n}|x^{n}\right)\geq 1-2^{-n\delta^{\prime }}.$$

**Lemma 4** ([30])**.** 
*Let $\left(x^{n},y^{n}\right)$ be emitted by the DMS $P_{XY}\left(\cdot ,\cdot \right)=W\left(\cdot |\cdot \right)P_{X}\left(\cdot \right)$ and let V∈W. For every ϵ>0, there exist a c(ϵ)>0 and $n_{0}\left(\epsilon \right)$ such that for $n\geq n_{0}\left(\epsilon \right)$ we have*
*1.* *$|\underset{V:∥V-W∥\leq \epsilon }{⋃}T_{V}^{n}\left(x^{n}\right)|\;\geq 2^{n(H\left(W|P_{X}\right)-c\left(\epsilon \right))}$,**2.* *$|\underset{V:∥V-W∥\leq \epsilon }{⋃}T_{V}^{n}\left(x^{n}\right)|\;\leq 2^{n(H\left(W|P_{X}\right)+c\left(\epsilon \right))}$,**3.* *If $∥V-W∥\leq \epsilon$, $T_{V}^{n}\left(x^{n}\right)\neq \emptyset$ and c(ϵ)→0 if ϵ→0, then*(95)$$|T_{V}^{n}\left(x^{n}\right)|\;\geq 2^{n(H\left(W|P_{X}\right)-c\left(\epsilon \right))}.$$

**Lemma 5** ([31])**.** 
*Let $\left\{X_{i}\right\}$ be i.i.d. RVs taking values in [0,1] with mean μ; then, ∀c>0 with p=μ+c≤1 we have*
(96)$$\mathrm{Pr}\left\{{\bar{X}}_{n}-\mu \geq c\right\}\leq \exp\left(-nD(p||\mu )\right)\leq \exp\left(-2nc^{2}\right),$$
*where ${\bar{X}}_{n}=\frac{1}{n}\sum_{i=1}^{n}X_{i}$.*

For arbitrary $D_{min}\leq D_{1}\leq D_{2}$, we define $X_{D_{1}}$ and $X_{D_{2}}$ by
(97)$$\begin{array}{cc} X_{D_{1}} & =\{x\in X,\;d^{★}\left(x\right)\leq D_{1}\}, \end{array}$$
(98)$$\begin{array}{cc} X_{D_{2}} & =\{x\in X,\;d^{★}\left(x\right)\leq D_{2}\}. \end{array}$$
It is clear that g(D) is a non-decreasing function, because $X_{D_{1}}\subseteq X_{D_{2}}$ for arbitrary $D_{1}\leq D_{2}$. Letting μ∈(0,1), we have
(99)$$\begin{array}{cc} g\left(\mu D_{1}+\left(1-\mu \right)D_{2}\right) & =\max_{x\in X_{\mu D_{1}+\left(1-\mu \right)D_{2}}}H\left(E[W_{S}\left(\cdot |x,S\right)]\right). \end{array}$$

It follows from Lemma 4 that $E_{S^{n}}\left[W_{S}^{n}\left(\cdot |{x^{★}}^{n},S^{n}\right)\right]$ is essentially uniform on $T^{n}$. Let the set $Y^{n}-T^{n}$ be denoted by $E^{★}$. Per Lemma 3, we have
(100)$$\begin{array}{cc} \; & \mathrm{Pr}\{Y^{n}\in E^{★}|X^{n}=\left({x^{★}}^{n},{x^{★}}^{n},\ldots ,{x^{★}}^{n}\right)\} \end{array}$$
(101)$$\begin{array}{cc} \; & =1-\mathrm{Pr}\{Y^{n}\notin T^{n}|X^{n}=\left({x^{★}}^{n},{x^{★}}^{n},\ldots ,{x^{★}}^{n}\right)\} \end{array}$$
(102)$$\begin{array}{cc} \; & \leq 2^{-n\delta^{\prime }}. \end{array}$$
As mentioned earlier, we have $|T^{n}|\approx 2^{nE_{S}\left[W_{S}\left(\cdot |{x^{★}}^{n},S^{n}\right)\right]}$. This quantity is the size of the correlated random experiment $\left(T^{n},E_{S^{n}}\left[W_{S}^{n}\left(\cdot |{x^{★}}^{n},S^{n}\right)\right]\right)$, which determines the growth of the ID rate. Let $C=\{\left(u_{j},D_{j}\right),\;j=1,\ldots ,M\}$ be an $\left(\left\lceil \sqrt{n}\right\rceil ,M,2^{-\sqrt{n}\delta }\right)$ code, where $u_{j}\in X_{D}^{\left\lceil \sqrt{n}\right\rceil }$ for each j=1,…,M. We concatenate the sequence ${x^{★}}^{n}=\left(x^{★},x^{★},\ldots ,x^{★}\right)\in X_{D}^{n}$ and the transmission code C to build an $\left(m,N,\lambda_{1},\lambda_{2}\right)$ ID feedback code $C^{\prime }=\left\{\left(f_{i},D_{i}^{\prime }\right),\;i=1,\ldots ,N\right\}$ for $W_{S}$. We now have $\lambda_{1},\lambda_{2}<\lambda ,\;\lambda \in \left(0,\frac{1}{2}\right)$. The concatenation is performed using the coloring functions $\left\{F_{i},\;i=1,\ldots ,N\right\}$. We choose a suitable set of coloring functions $\left\{F_{i},\;i=1,\cdots ,N\right\}$ at random. Every coloring function $F_{i}:T^{n}⟶\left\{1,\ldots ,M\right\}$ corresponds to an ID message *i* and maps each element $y^{n}\in T^{n}$ to an element $F_{i}\left(y^{n}\right)$ in a smaller set {1,…,M}. After $y^{n}\in T^{n}$ has been received by the sender (encoder and estimator) via the noiseless feedback channel, if i∈{1,…,N} is available, then the encoder sends $u_{F_{i}\left(y^{n}\right))}$. Note that we define an encoding strategy $f_{i}=\left[f_{i}^{1},\ldots ,f_{i}^{m}\right]\in F_{m}$ for each coloring function $F_{i}$, as presented in Definition 4. If $y^{n}\notin T^{n}$, then an error is declared. This error probability goes to zero as *n* goes to infinity, as computed in (102). For a fixed family of maps $\left\{F_{i},\;i=1,\ldots ,N\right\}$ and for each i∈{1,…,N}, we define the decoding sets $D\left(F_{i}\right)={⋃}_{y^{n}\in T^{n}}\left\{y^{n}\right\}\times D_{F_{i}\left(y^{n}\right)}$.

#### 3.5.3. Error Analysis

Next, we analyze the maximal error performance of the deterministic ID feedback code. For our analysis of the error of the first kind, we choose a fixed set $\left\{F_{i},\;i=1,\ldots ,N\right\}$. The error of the first kind is upper-bounded by
(103)$$\begin{array}{cc} \; & E_{S^{m}}\left[W_{S}^{m}\left({D^{\prime }}_{i}^{c}|f_{i},S^{m}\right)\right] \end{array}$$
(104)$$\begin{array}{cc} & =E_{S^{m}}\left[W_{S}^{m}\left({D(F_{i})}^{c}|f_{i},S^{m}\right)\right] \end{array}$$
(105)$$\begin{array}{cc} \; & \overset{\left(a\right)}{\leq }\sum_{s^{m}\in S^{m}}P_{S}^{m}\left(s^{m}\right)\left(W_{S}^{n}\left({\left(T^{n}\right)}^{c}|{x^{★}}^{n},s^{n}\right)+W_{S}^{\left\lceil \sqrt{n}\right\rceil }\left(D_{F_{i}\left(y^{n}\right)}|u_{F_{i}\left(y^{n}\right)},s^{\left\lceil \sqrt{n}\right\rceil }\right)\right) \end{array}$$
(106)$$\begin{array}{cc} \; & \overset{\left(b\right)}{\leq }2^{-n\delta }+2^{-\sqrt{n}\delta^{\prime }}, \end{array}$$
where (a) follows from the memorylessness property of the channel and the union bound, while (b) follows from Lemma 3 and the definition of the transmission code C.

In order to achieve a small error of the second kind, we choose suitable maps $\left\{F_{i},\;i=1,\ldots ,N\right\}$ randomly. For i∈{1,…,N}, $y^{n}\in T^{n}$, let ${\bar{F}}_{i}\left(y^{n}\right)$ be independent RVs such that
(107)$$\mathrm{Pr}\left\{{\bar{F}}_{i}\left(y^{n}\right)=j\right\}=\frac{1}{M},\;j\in \left\{1,\ldots ,M\right\}.$$
Let $F_{1}$ be a realization of ${\bar{F}}_{1}$. For each $y^{n}\in T^{n}$, we define the RVs $\psi_{y^{n}}=\psi_{y^{n}}\left({\bar{F}}_{2}\right)$ analogously to Section IV of [14]:
(108)$$\psi_{y^{n}}=\psi_{y^{n}}\left({\bar{F}}_{2}\right)=\left\{\begin{array}{cc} 1, & \mathrm{if}\;F_{1}\left(y^{n}\right)={\bar{F}}_{2}\left(y^{n}\right)\\ 0, & \mathrm{otherwise}. \end{array}\right.$$The $\psi_{y^{n}}$ are also independent for every $y^{n}\in T^{n}$. The expectation of $\psi_{y^{n}}$ is computed as follows:
(109)$$\begin{array}{cc} E[\psi_{y^{n}}] & =\mathrm{Pr}\left\{F_{1}\left(y^{n}\right)={\bar{F}}_{2}\left(y^{n}\right)\right\}=\frac{1}{M}. \end{array}$$
Because the $\psi_{y^{n}}$ are i.i.d. for all $y^{n}\in T^{n}$, we can apply Hoeffding’s inequality Lemma 5 to obtain the following Lemma.

**Lemma 6.** 
*For λ∈(0,1), $\frac{1}{M}<\lambda$ for each channel $V_{S}^{a}$ with $∥V_{S}^{a}-W_{S}^{a}∥\leq \epsilon$, while for each $n\geq n_{0}\left(\epsilon \right)$ we have*

(110)
$$\begin{array}{c} \mathrm{Pr}\{\sum_{y^{n}\in T^{n}}\psi_{y^{n}}>|T_{V_{S}^{a}}^{n}\left({x^{★}}^{n}\right)\left|\cdot \lambda \right\}\leq 2^{-|T^{n}|\cdot \lambda \sqrt{n}\epsilon }. \end{array}$$



We can derive an upper bound on the error of the second kind for those values of ${\bar{F}}_{2}$ satisfying Lemma 6:
(111)$$\begin{array}{cc} \; & E_{S^{m}}\left[W_{S}^{m}\left(D\left({\bar{F}}_{2}\right)|f_{1},S^{m}\right)\right] \end{array}$$
(112)$$\begin{array}{cc} \; & =\sum_{s^{m}\in S^{m}}P_{S}^{m}\left(s^{m}\right)W_{S}^{m}\left(D\left({\bar{F}}_{2}\right)|f_{1},s^{m}\right) \end{array}$$
(113)$$\begin{array}{cc} \; & =\sum_{s^{m}\in S^{m}}P_{S}^{m}\left(s^{m}\right)W_{S}^{m}\left(D\left({\bar{F}}_{2}\right)\cap \left(\left(T^{n}\times Y^{\left\lceil \sqrt{n}\right\rceil }\right)\cup {\left(T^{n}\times Y^{\left\lceil \sqrt{n}\right\rceil }\right)}^{c}\right)|f_{1},s^{m}\right) \end{array}$$
(114)$$\begin{array}{cc} \; & \overset{\left(a\right)}{\leq }\sum_{s^{m}\in S^{m}}P_{S}^{m}\left(s^{m}\right)W_{S}^{m}\left({\left(T^{n}\times Y^{\left\lceil \sqrt{n}\right\rceil }\right)}^{c}|f_{1},s^{m}\right) \end{array}$$
(115)$$\begin{array}{cc} \; & \;+\sum_{s^{m}\in S^{m}}P_{S}^{m}\left(s^{m}\right)W_{S}^{m}\left(D\left({\bar{F}}_{2}\right)\cap \left(T^{n}\times Y^{\left\lceil \sqrt{n}\right\rceil }\right)|f_{1},s^{m}\right) \end{array}$$
(116)$$\begin{array}{cc} \; & \overset{\left(b\right)}{\leq }\sum_{s^{n}\in S^{n}}P_{S}^{n}\left(s^{n}\right)W_{S}^{n}\left({\left(T^{n}\right)}^{c}|{x^{★}}^{n},s^{n}\right) \end{array}$$
(117)$$\begin{array}{cc} \; & +\sum_{s^{m}\in S^{m}}P_{S}^{m}\left(s^{m}\right)(\sum_{\begin{array}{c} y^{n}\in T^{n}\\ F_{1}\left(y^{n}\right)\neq {\bar{F}}_{2}\left(y^{n}\right) \end{array}}W_{S}^{n}\left(y^{n}|{x^{★}}^{n},s^{n}\right)\cdot W_{S}^{\left\lceil \sqrt{n}\right\rceil }\left(y^{\left\lceil \sqrt{n}\right\rceil }|u_{F_{1}\left(y^{n}\right)},s^{\left\lceil \sqrt{n}\right\rceil }\right) \end{array}$$
(118)$$\begin{array}{cc} \; & \;+\sum_{\begin{array}{c} y^{n}\in T^{n}\\ F_{1}\left(y^{n}\right)={\bar{F}}_{2}\left(y^{n}\right) \end{array}}W_{S}^{n}\left(y^{n}|{x^{★}}^{n},s^{n}\right)) \end{array}$$
(119)$$\begin{array}{cc} \; & \leq 2^{-n\delta^{\prime }}+2^{-\sqrt{n}\delta }+\sum_{s^{n}\in S^{n}}P_{S}^{n}\left(s^{n}\right)\sum_{\begin{array}{c} y^{n}\in T^{n}\\ F_{1}\left(y^{n}\right)={\bar{F}}_{2}\left(y^{n}\right) \end{array}}W_{S}^{n}\left(y^{n}|{x^{★}}^{n},s^{n}\right) \end{array}$$
(120)$$\begin{array}{cc} \; & \overset{\left(c\right)}{\leq }2^{-n\delta^{\prime }}+2^{-\sqrt{n}\delta } \end{array}$$
(121)$$\begin{array}{cc} \; & \;+\sum_{V_{S}^{a}:\left|V_{S}^{a}-W_{S}^{a}\right|\leq \epsilon }{\left(W_{S}^{a}\right)}^{n}\left(T_{V_{S}^{a}}^{n}\left({x^{★}}^{n}\right)|{x^{★}}^{n}\right)\cdot \left(\sum_{y^{n}\in T_{V_{S}^{a}}^{n}\left({x^{★}}^{n}\right)}\psi_{y^{n}}\right)\cdot {\left|T_{V_{S}^{a}}^{n}\left({x^{★}}^{n}\right)\right|}^{-1} \end{array}$$
(122)$$\begin{array}{cc} \; & \overset{\left(d\right)}{\leq }2^{-n\delta^{\prime }}+2^{-\sqrt{n}\delta }+\lambda \end{array}$$
where (a) follows from the union bound, (b) follows from the memorylessness property of the channel and the union bound, (c) follows from Lemma 3 along with the definition of the transmission code C and the definition of the set $T^{n}$ in (90), and (d) follows from Lemma 6.

We repeatedly perform the same analysis of the error of the second kind for all pairs $\left(i_{1},i_{2}\right)\in {\left\{1,\ldots ,N\right\}}^{2},\;i_{1}\neq i_{2}$. For simplicity of notation, we denote the error of the second kind between the pair $\left(i_{1},i_{2}\right)$ by $\mu_{2}^{\left(i_{1},i_{2}\right)}$. We have
(123)$$\begin{array}{cc} \; & \mathrm{Pr}\{C^{\prime }\;\mathrm{is}\;\mathrm{not}\;\mathrm{an}\;\left(n,N,\lambda_{1},\lambda_{2}\right)\;\mathrm{code}\} \end{array}$$
(124)$$\begin{array}{cc} & =\mathrm{Pr}\{\underset{\begin{array}{c} i_{1},i_{2}\in \left\{1,\ldots ,N\right\}\\ i_{1}\neq i_{2} \end{array}\mu_{2}^{\left(i_{1},i_{2}\right)}}{⋃}\geq \lambda_{2}\} \end{array}$$
(125)$$\begin{array}{cc} \; & =\mathrm{Pr}\{\underset{\begin{array}{c} i_{1},i_{2}\in \left\{1,\ldots ,N\right\}\\ i_{1}\neq i_{2} \end{array}}{⋃}\mu_{2}^{\left(i_{1},i_{2}\right)}\geq \lambda +2^{-n\delta^{\prime }}+2^{-\sqrt{n}\delta }\} \end{array}$$
(126)$$\begin{array}{cc} \; & \overset{\left(a\right)}{\leq }N\cdot \left(N-1\right)\cdot 2^{-|T^{n}|\cdot \lambda \sqrt{n}\epsilon }, \end{array}$$
where (a) follows from the union bound, Equation (122) and Lemma 6. It is verifiable that we can construct an ID feedback code for $W_{S}$ with cardinality *N* satisfying
(127)$$N\geq {\left(n+1\right)}^{-2|X||Y|}\cdot 2^{|T^{n}|\cdot \lambda \sqrt{n}\epsilon }$$
and with an error of the second kind upper-bounded as in (122).

The next step in the proof is dedicated to the state estimator. The per-symbol distortion defined in (15) can be rewritten as follows:
(128)$$\begin{array}{cc} d_{t}\; & =E\left[d(S_{t},{\hat{S}}_{t})\right] \end{array}$$
(129)$$\begin{array}{cc} \; & =E\left[E\left[d\left(S_{t},{\hat{S}}_{t}\right)|X_{t},Y_{t}\right]\right] \end{array}$$
(130)$$\begin{array}{cc} \; & =\sum_{(x,y)\in X\times Y}P_{X_{t}Y_{t}}\left(x,y\right)\sum_{s_{t}\in S}P_{S_{t}|X_{t}Y_{t}}\left(s|x,y\right)\sum_{{\hat{s}}_{t}\in S}P_{{\hat{S}}_{t}|X_{t}Y_{t}}\left(\hat{s}|x,y\right)d\left(s,\hat{s}\right) \end{array}$$
(131)$$\begin{array}{cc} \; & =\sum_{(x,y)\in X\times Y}P_{X_{t}Y_{t}}\left(x,y\right)\sum_{s_{t}\in S}P_{S_{t}|X_{t}Y_{t}}\left(s|x,y\right)d\left(s,h^{★}\left(x,y\right)\right) \end{array}$$
(132)$$\begin{array}{cc} \; & =\sum_{x\in X}P_{X_{t}}\left(x\right)E_{S_{t}Y_{t}}\left[d\left(S_{t},h^{★}\left(X_{t},Y_{t}\right)\right)|X_{t}=x\right] \end{array}$$
(133)$$\begin{array}{cc} \; & =\sum_{x\in X}P_{X_{t}}\left(x\right)d^{★}\left(x\right) \end{array}$$
(134)$$\begin{array}{cc} \; & \leq D. \end{array}$$
This completes the direct proof of Theorem 4.

### 3.6. Converse Proof of Theorem 4

**Proof.** For the converse proof, we use the techniques for deterministic ID for $W_{S}$ as described in Section 3.2. We first extend Lemma 1 to the joint deterministic ID and sensing problem. □

**Lemma 7.** 
*For any feedback strategy $f_{D}=\left[f_{D}^{1},f_{D}^{2},\cdots ,f_{D}^{n}\right]$ which satisfies the per-symbol distortion constraint as described in (17), i.e., for all $t\in \left\{1,2,\cdots ,n\right\}$, $d^{★}\left(f_{D}^{t}\right)\leq D$ and for any $\mu \in \left(0,1\right)$, we have*

(135)
$$\begin{array}{c} \min_{E_{3}\subset Y^{n}:E_{S^{n}}\left[W_{S}^{n}\left(E_{3}|f_{D},S^{n}\right)\right]\geq 1-\mu }\left|E_{3}\right|\leq K_{3}, \end{array}$$

*where $K_{3}$ is provided by*

(136)
$$\begin{array}{c} K_{3}=2^{n\max_{x\in X_{D}}H\left(E\left[W_{S}\left(\cdot |x\right)\right]\right)+\alpha \sqrt{n}}=2^{nH\left(E\left[W_{S}\left(\cdot |x^{★}\right)\right]\right)+\alpha \sqrt{n}}, \end{array}$$

*where*

(137)
$$\begin{array}{c} \alpha =\sqrt{\frac{\beta }{\mu }},\;\beta =\max\left(\log^{2}\left(3\right),\log^{2}\left(\left|Y\right|\right)\right). \end{array}$$



**Proof.** Let $E_{3}^{★}\subset Y^{n}$ be defined as follows:
(138)$$\begin{array}{c} E_{3}^{★}=\left\{y^{n}\in Y^{n},-\log E_{S^{n}}\left[W_{S}^{n}\left(f_{D},S^{n}\right)\right]\leq K_{3}\right\}. \end{array}$$
Define an RV $Z_{t}=-\log E_{S_{t}}\left[W_{S}\left(Y_{t}|f_{D}^{t}\left(Y^{t-1}\right),S_{t}\right)\right]$. Per (40), we have
(139)$$\begin{array}{c} E_{S^{n}}\left[W_{S}^{n}\left(E_{3}^{★}|f_{D},S^{n}\right)\right]=Pr\left\{\sum_{t=1}^{n}Z_{t}\leq \log K_{3}\right\}. \end{array}$$
Similarly, for all $t\in \left\{1,2,\cdots ,n\right\}$, we define an RV $U_{t}=Z_{t}-E\left[Z_{t}|Y^{t-1}\right]$. It has been shown in (44) and (48) that $E\left[U_{t}|Y^{t-1}\right]=0$ and $E\left[U_{t}\right]=0$.Moreover, for all $t\in \left\{1,2,\cdots ,n\right\}$ and for all $y^{t-1}\in Y^{t-1}$, we have
(140)$$\begin{array}{cc} E\left[Z_{t}|y^{t-1}\right]\; & =\sum_{y_{t}\in Y}\left(-E_{S_{t}}\left[W_{S}\left(Y_{t}|f_{D}^{t}\left(Y^{t-1}\right),S_{t}\right)\right]\log E_{S_{t}}\left[W_{S}\left(Y_{t}|f_{D}^{t}\left(Y^{t-1}\right),S_{t}\right)\right]\right) \end{array}$$
(141)$$\begin{array}{cc} \; & \leq \max_{x\in X_{D}}H\left(E_{S}\left[W_{S}\left(\cdot |x,S\right)\right]\right) \end{array}$$
(142)$$\begin{array}{cc} \; & =H\left(E_{S}\left[W_{S}\left(\cdot |x^{★},S\right)\right]\right). \end{array}$$
By combining (139) and (142), we have
(143)$$\begin{array}{cc} E_{S^{n}}\left[W_{S}^{n}\left(f_{D},S^{n}\right)\right]\; & \geq Pr\left\{\sum_{t=1}^{n}U_{t}\leq \alpha \sqrt{n}\right\} \end{array}$$
(144)$$\begin{array}{cc} \; & \geq 1-\mu . \end{array}$$
This completes the proof of Lemma 7 □

The subsequent steps in the proof are identical to Section 3.2.

### 3.7. Direct Proof of Theorem 5

**Proof.** For the direct proof of this theorem, we follow a code construction similar to that presented in [14], with one key difference, namely, that we optimize only over input distributions that satisfy the per-symbol constraint $d^{★}\left(P\right)\leq D$. □

#### 3.7.1. Coding Scheme

We construct a randomized ID code with blocklength $m=n+\sqrt{n}$ by concatenating two transmission codes, which is described in detail later. The first *n* symbols are allocated for generation of common randomness. We employ a distribution
$$P^{*}=\arg\max_{P\in P(X)}H(\sum_{x\in X}P\left(x\right)E\left[W_{S}\left(\cdot |x,S\right))\right],$$
where the maximization is performed over distributions subject to the constraint $P_{D}=\left\{P\in P\left(X\right)|d^{★}\left(P\right)\leq D\right\}$. Regardless of which identity i∈{1,…,N} we want to identify, the sender first sends *n* symbols with respect to the distribution ${P^{★}}^{n}\in P_{D}^{n}$ over the state-dependent channel $W_{S}$. The received sequence $y^{n}\in Y^{n}$ becomes known to the sender (estimator and encoder) via the noiseless feedback link. The feedback provides the sender and the receiver with knowledge of the outcome of the correlated random experiment $\left(Y^{n},\sum_{x^{n}\in X^{n}}P^{★n}\left(x^{n}\right)E_{S^{n}}\left[W_{S}^{n}\left(\cdot |x^{n},S^{n}\right)\right]\right)$.

#### 3.7.2. Common Randomness Generation

Similar to the deterministic coding scheme, we want to generate uniform common randomness. Therefore, we convert our correlated random experiment
$$\left(Y^{n},\sum_{x^{n}\in X^{n}}P^{★n}\left(x^{n}\right)E_{S^{n}}\left[W_{S}^{n}\left(\cdot |{x^{★}}^{n},S^{n}\right)\right]\right)$$
to a uniform one $\left(T^{\prime n},\sum_{x^{n}\in X^{n}}P^{★n}\left(x^{n}\right)E_{S^{n}}\left[W_{S}^{n}\left(\cdot |{x^{★}}^{n},S^{n}\right)\right]\right))$. For ϵ>0, the set $T^{\prime n}$ is provided by
(145)$$T^{\prime n}=\underset{V_{S}^{a}:∥V_{S}^{a}-W_{S}^{a}∥\leq \epsilon }{⋃}\sum_{x\in X}P^{★n}\left(x^{n}\right)T_{V_{S}^{a}}^{n}\left(x^{n}\right).$$

Per Lemma 4, we can obtain the following corollary.

**Corollary 1.** 
*Let $\left(x^{n},y^{n}\right)$ be emitted by the DMS $P_{XY}\left(\cdot ,\cdot \right)=W\left(\cdot |\cdot \right)P_{X}\left(\cdot \right)$ and let V∈W. For every ϵ>0, there exist a c(ϵ)>0 and $n_{0}\left(\epsilon \right)$ such that for $n\geq n_{0}\left(\epsilon \right)$ we have:*
*1.* 
*$|\underset{V:∥V-W∥\leq \epsilon }{⋃}T_{V}^{n}\left(x^{n}\right)|\;\geq 2^{n(H\left(\sum_{x\in X}P_{X}\left(x\right)W\left(\cdot |x\right)\right)-c\left(\epsilon \right))}$,*
*2.* 
*$|\underset{V:∥V-W∥\leq \epsilon }{⋃}T_{V}^{n}\left(x^{n}\right)|\;\leq 2^{n(H\left(\sum_{x\in X}P_{X}\left(x\right)W\left(\cdot |x\right)\right)+c\left(\epsilon \right))}$,*
*3.* 
*If $∥V-W∥\leq \epsilon$, $T_{V}^{n}\left(x^{n}\right)\neq \emptyset$ and c(ϵ)→0 if ϵ→0, then*

(146)
$$|T_{V}^{n}\left(x^{n}\right)|\;\geq 2^{n(H\left(\sum_{x\in X}P_{X}\left(x\right)W\left(\cdot |x\right)\right)-c\left(\epsilon \right))}.$$




Therefore, we have $\left|T^{\prime n}\right|\approx 2^{nH\left(\sum_{x\in X}P\left(x\right)E_{S}\left[W(\cdot |x,S)\right]\right)}$. The sender generates randomness according to the random experiment $\left(T^{\prime n},\sum_{x^{n}\in X^{n}}P^{★n}\left(x^{n}\right)E_{S^{n}}\left[W_{S}^{n}\left(\cdot |x^{n},S^{n}\right)\right]\right)$. Asymptotically, the error probability of $y^{n}\notin T^{\prime n}$ goes to zero. Similar to the deterministic scheme, we prepare the coloring functions $\left\{F_{i},i=1,\cdots ,N\right\}$. The last $\left\lceil \sqrt{n}\right\rceil$ symbols are used to transmit $F_{i}\left(y^{n}\right)$ using a standard $\left(\left\lceil \sqrt{n}\right\rceil ,M,2^{-n\sqrt{n}\delta }\right)$ transmission code $C=\left\{\left(u_{j},D_{j}\right),j=1,\cdots ,M\right\}$, where $u_{j}\in X_{D}^{\left\lceil \sqrt{n}\right\rceil }$ for each j=1,⋯,M. The probability distribution for encoding is defined as $Q_{F}\left(\cdot |i\right)={P^{★}}^{n}\times I\left\{x_{n}^{m}=F_{i}\left(y^{n}\right)\right\}$ and the decoding region is provided by $D\left(F_{i}\right)={⋃}_{y^{n}\in T^{\prime }}\left\{y^{n}\right\}\times D\left(F_{i}\left(y^{n}\right)\right)$.

#### 3.7.3. Error Analysis

Subsequently, for all i=1,⋯,N, the error of the first kind $P_{e,1}\left(i\right)$ is upper-bounded by
(147)$$\begin{array}{cc} \; & P_{e,1}\left(i\right) \end{array}$$
(148)$$\begin{array}{cc} \; & =\sum_{f\in F^{m}}Q_{F}\left(f|i\right)E_{S^{m}}\left[W_{S}^{m}\left(D\left(F_{i}\right)|f,S^{m}\right)\right] \end{array}$$
(149)$$\begin{array}{cc} \; & =\sum_{s^{m}\in S^{m}}P_{S}^{m}\left(s^{m}\right)\sum_{f\in F^{m}}Q_{F}\left(f|i\right)W_{S}^{m}\left({\left(\underset{y^{n}\in T^{\prime }}{⋃}y^{n}\times D_{F_{i}\left(y^{n}\right)}\right)}^{c}|f,S^{m}\right) \end{array}$$
(150)$$\begin{array}{cc} \; & \overset{\left(a\right)}{\leq }\sum_{s^{m}\in S^{m}}P_{S}^{m}\left(s^{m}\right)\left(\sum_{x^{n}\in X^{n}}{P^{★}}^{n}\left(x^{n}\right)W_{S}^{n}\left(T^{\prime c}|x^{n},S^{n}\right)+W_{S}^{\left\lceil \sqrt{n}\right\rceil }\left(D_{F_{i}\left(y^{n}\right)}^{c}|u_{F_{i}\left(y^{n}\right)},S^{\left\lceil \sqrt{n}\right\rceil }\right)\right) \end{array}$$
(151)$$\begin{array}{cc} \; & \overset{\left(b\right)}{\leq }2^{-n\delta }+2^{-\sqrt{n}\delta^{\prime }}, \end{array}$$
where (a) follows from the union bound and (b) follows from Corollary 1. Furthermore, for all i,j=1,⋯,N with i≠j, the probability of the error of the second kind $P_{e,2}\left(i,j\right)$ should be asymptotically upper-bounded by λ. Without loss of generality, we fix i=1, j=2 and examine the error probability $P_{e,2}\left(1,2\right)$:
(152)$$\begin{array}{cc} \; & P_{e,2}\left(1,2\right) \end{array}$$
(153)$$\begin{array}{cc} \; & =\sum_{f\in F^{m}}Q\left(f|i=1\right)E_{S^{m}}\left[W_{S}^{m}\left(D\left({\bar{F}}_{2}\right)|f,S^{m}\right)\right] \end{array}$$
(154)$$\begin{array}{cc} \; & =\sum_{s^{m}\in S^{m}}P_{S}^{m}\left(s^{m}\right)\sum_{f\in F^{m}}Q\left(f|i=1\right)W_{S}^{m}\left(D\left({\bar{F}}_{2}\right)|f,s^{m}\right) \end{array}$$
(155)$$\begin{array}{cc} \; & =\sum_{s^{m}\in S^{m}}P_{S}^{m}\left(s^{m}\right)\sum_{f\in F^{m}}Q\left(f|i=1\right)\cdot \end{array}$$
(156)$$\begin{array}{cc} \; & \;\cdot W_{S}^{m}\left(D\left({\bar{F}}_{2}\right)\cap \left(\left(T^{\prime n}\times Y^{\left\lceil \sqrt{n}\right\rceil }\right)\cup {\left(T^{\prime n}\times Y^{\left\lceil \sqrt{n}\right\rceil }\right)}^{c}\right)|f,s^{m}\right) \end{array}$$
(157)$$\begin{array}{cc} \; & \overset{\left(a\right)}{\leq }\sum_{s^{m}\in S^{m}}P_{S}^{m}\left(s^{m}\right)\sum_{f\in F^{m}}Q\left(f|i=1\right)W_{S}^{m}\left({\left(T^{n}\times Y^{\left\lceil \sqrt{n}\right\rceil }\right)}^{c}|f,s^{m}\right) \end{array}$$
(158)$$\begin{array}{cc} \; & \;+\sum_{s^{m}\in S^{m}}P_{S}^{m}\left(s^{m}\right)\sum_{f\in F^{m}}Q\left(f|i=1\right)W_{S}^{m}\left(D\left({\bar{F}}_{2}\right)\cap \left(T^{n}\times Y^{\left\lceil \sqrt{n}\right\rceil }\right)|f,s^{m}\right) \end{array}$$
(159)$$\begin{array}{cc} \; & \overset{\left(b\right)}{\leq }\sum_{s^{m}\in S^{m}}P_{S}^{m}\left(s^{m}\right)\sum_{x^{n}\in X^{n}}{P^{★}}^{n}\left(x^{n}\right)W_{S}^{n}\left({\left(T^{n}\right)}^{c}|x^{n},s^{n}\right)W_{S}^{\left\lceil \sqrt{n}\right\rceil }\left(D_{F_{i}\left(y^{n}\right)}^{c}|u_{F_{i}\left(y^{n}\right)},s^{\left\lceil \sqrt{n}\right\rceil }\right) \end{array}$$
(160)$$\begin{array}{cc} \; & +\sum_{s^{m}\in S^{m}}P_{S}^{m}\left(s^{m}\right)(\sum_{\begin{array}{c} y^{n}\in T^{n}\\ F_{1}\left(y^{n}\right)\neq {\bar{F}}_{2}\left(y^{n}\right) \end{array}}\sum_{x^{n}\in X^{n}}{P^{★}}^{n}\left(x^{n}\right)W_{S}^{n}\left(y^{n}|x^{n},s^{n}\right)\cdot \end{array}$$
(161)$$\begin{array}{cc} \; & \;\cdot W_{S}^{\left\lceil \sqrt{n}\right\rceil }\left(y^{\left\lceil \sqrt{n}\right\rceil }|u_{F_{1}\left(y^{n}\right)},s^{\left\lceil \sqrt{n}\right\rceil }\right)+\sum_{\begin{array}{c} y^{n}\in T^{n}\\ F_{1}\left(y^{n}\right)={\bar{F}}_{2}\left(y^{n}\right) \end{array}}\sum_{x^{n}\in X^{n}}{P^{★}}^{n}\left(x^{n}\right)W_{S}^{n}\left(y^{n}|x^{n},s^{n}\right)) \end{array}$$
(162)$$\begin{array}{cc} \; & \leq 2^{-n\delta^{\prime }}+2^{-\sqrt{n}\delta }+\sum_{s^{n}\in S^{n}}P_{S}^{n}\left(s^{n}\right)\sum_{\begin{array}{c} y^{n}\in T^{n}\\ F_{1}\left(y^{n}\right)={\bar{F}}_{2}\left(y^{n}\right) \end{array}}\sum_{x^{n}\in X^{n}}{P^{★}}^{n}\left(x^{n}\right)W_{S}^{n}\left(y^{n}|x^{n},s^{n}\right) \end{array}$$
(163)$$\begin{array}{cc} \; & \overset{\left(c\right)}{\leq }2^{-n\delta^{\prime }}+2^{+}\sum_{x^{n}\in X^{n}}{P^{★}}^{n}\left(x^{n}\right)\sum_{V_{S}^{a}:\left|V_{S}^{a}-W_{S}^{a}\right|\leq \epsilon }{\left(W_{S}^{a}\right)}^{n}\left({T^{\prime }}_{n}^{V_{S}^{a}}\left(x^{n}\right)|x^{n}\right)\cdot \end{array}$$
(164)$$\begin{array}{cc} \; & \;\cdot \left(\sum_{y^{n}\in {T^{\prime }}_{n}^{V_{S}^{a}}\left(x^{n}\right)}\psi_{y^{n}}\right)\cdot {\left|{T^{\prime }}_{n}^{V_{S}^{a}}\left(x^{n}\right)\right|}^{-1} \end{array}$$
(165)$$\begin{array}{cc} \; & \overset{\left(d\right)}{\leq }2^{-n\delta^{\prime }}+2^{-\sqrt{n}\delta }+\lambda \end{array}$$
where (a) follows from the union bound, (b) follows from the memoryless channel and the union bound, (c) follows from Lemma 3 along with the definition of the transmission code C and the definition of the set $T^{n}$ in (90), and (d) follows from Corollary 1.

We repeatedly perform the same analysis of the error of the second kind for all pairs $\left(i_{1},i_{2}\right)\in {\left\{1,\ldots ,N\right\}}^{2},\;i_{1}\neq i_{2}$. It is verifiable that we can construct a randomized ID feedback code for $W_{S}$ with cardinality *N* satisfying
(166)$$N\geq {\left(n+1\right)}^{-2|X||Y|}\cdot 2^{|T^{\prime n}|\cdot \lambda \sqrt{n}\epsilon }$$
and with errors of the first and second kind that are upper-bounded as in (11) and (12), respectively.

Finally the state estimator is checked. The per-symbol distortion defined in (15) can be rewritten as follows:
(167)$$\begin{array}{cc} d_{t}\; & =E\left[d(S_{t},{\hat{S}}_{t})\right] \end{array}$$
(168)$$\begin{array}{cc} \; & =E\left[E\left[d\left(S_{t},{\hat{S}}_{t}\right)|X_{t},Y_{t}\right]\right] \end{array}$$
(169)$$\begin{array}{cc} \; & =\sum_{(x,y)\in X\times Y}P_{X_{t}Y_{t}}\left(x,y\right)\sum_{s_{t}\in S}P_{S_{t}|X_{t}Y_{t}}\left(s|x,y\right)\sum_{{\hat{s}}_{t}\in S}P_{{\hat{S}}_{t}|X_{t}Y_{t}}\left(\hat{s}|x,y\right)d\left(s,\hat{s}\right) \end{array}$$
(170)$$\begin{array}{cc} \; & =\sum_{(x,y)\in X\times Y}P_{X_{t}Y_{t}}\left(x,y\right)\sum_{s_{t}\in S}P_{S_{t}|X_{t}Y_{t}}\left(s|x,y\right)d\left(s,h^{★}\left(x,y\right)\right) \end{array}$$
(171)$$\begin{array}{cc} \; & =\sum_{x\in X}P_{X_{t}}\left(x\right)E_{S_{t}Y_{t}}\left[d\left(S_{t},h^{★}\left(X_{t},Y_{t}\right)\right)|X_{t}=x\right] \end{array}$$
(172)$$\begin{array}{cc} \; & =d^{★}\left(P\right) \end{array}$$
(173)$$\begin{array}{cc} \; & \leq D. \end{array}$$
This completes the direct proof of Theorem 5. 

### 3.8. Converse Proof of Theorem 5

**Proof.** We first extend Lemma 2 to the joint randomized ID and sensing problem. □

**Lemma 8.** 
*For any randomized feedback strategy $Q_{D}\left(f\right)=\prod_{t=1}^{n}Q_{D}^{t}\left(f^{t}\right)$ over all n-length feedback encoding sets $F_{n}$ which satisfies the per symbol distortion constraint as described in (142), i.e., for all t∈{1,2,⋯,n}, $d^{★}\left(Q_{D}^{t}\left(f^{t}\right)\right)\leq D$ and for any μ∈(0,1), we have*

(174)
$$\begin{array}{c} \min_{E_{4}\subset Y^{n}:\sum_{f\in F_{n}}Q_{D}\left(f\right)E_{S^{n}}\left[W_{S}^{n}\left(E_{4}|f,S^{n}\right)\right]\geq 1-\mu }\left|E_{4}\right|\leq K_{4}, \end{array}$$

*where $K_{4}$ is provided by*

(175)
$$\begin{array}{c} K_{4}=2^{n\max_{P\in P^{D}}H\left(\sum_{x\in X}P\left(x\right)E\left[W_{S}\left(\cdot |x,S\right)\right]\right)+\alpha \sqrt{n}}=2^{nH\left(\sum_{x\in X}P^{★}\left(x\right)E\left[W_{S}\left(\cdot |x,S\right)\right]\right)+\alpha \sqrt{n}}, \end{array}$$

*where α=βμ, β=maxlog23,log2|Y|.*


**Proof.** Define a set $E_{4}^{★}\subset$ as follows:
(176)$$\begin{array}{c} E_{4}^{★}=\left\{y^{n}\in Y^{n},-\log\sum_{f\in F_{n}}Q_{D}\left(f\right)E_{S^{n}}\left[W_{S}\left(y^{n}|f,S^{n}\right)\right]\leq \log K_{4}\right\}. \end{array}$$
Define an auxiliary RV $Z_{t}$ as follows:
(177)$$Z_{t}=-\log\left(\sum_{f^{t}\in F^{t}}Q_{D}^{t}\left(f^{t}\right)E_{S_{t}}\left[W_{S}^{n}\left(Y_{t}|f^{t}\left(Y^{t-1}\right),S_{t}\right)\right]\right).$$
Then, per (80), we have
(178)$$\begin{array}{c} \sum_{f\in F_{n}}Q_{D}\left(f\right)E\left[W_{S}^{n}\left(E_{4}|f,S^{n}\right)\right]=Pr\left\{\sum_{t=1}^{n}Z_{t}\leq \log K_{4}\right\}. \end{array}$$
Similarly, for any $y^{t-1}\in Y^{t-1}$, we can examine
(179)$$\begin{array}{cc} \; & E\left[Z_{t}|y^{t-1}\right] \end{array}$$
(180)$$\begin{array}{cc} \; & =\sum_{y_{t}\in Y}\left(-\sum_{f^{t}\in F^{t}}Q_{D}^{t}\left(f^{t}\right)E_{S_{t}}\left[W_{S}^{n}\left(y_{t}|f^{t}\left(y^{t-1}\right),S_{t}\right)\right]\log\left(\sum_{f^{t}\in F^{t}}Q_{D}^{t}\left(f^{t}\right)E_{S_{t}}\left[W_{S}^{n}\left(y_{t}|f^{t}\left(y^{t-1}\right),S_{t}\right)\right]\right)\right) \end{array}$$
(181)$$\begin{array}{cc} \; & \leq \max_{P\in P_{D}}H\left(\sum_{x\in X}P\left(x\right)E\left[W_{S}\left(\cdot |x,S\right)\right]\right) \end{array}$$
(182)$$\begin{array}{cc} \; & =H\left(\sum_{x\in X}P^{★}\left(x\right)E\left[W_{S}^{n}\left(\cdot \right|x,S\right]\right). \end{array}$$
By applying Chebyshev’s inequality, we complete the proof of Lemma 8. □

The subsequent steps in the proof are the same as in Section 3.6.

## 4. Average Distortion

In addition to the per-symbol distortion constraint, an alternative and more flexible distortion constraint is the average distortion. This approach is valuable because it relaxes the per-symbol fidelity requirement, allowing for minor variations in individual symbol quality as long as the overall average distortion remains below a specified threshold. As defined in [32], the average distortion for a sequence of symbols is provided by
(183)$$\begin{array}{cc} {\bar{d}}^{n} & =E_{S^{n}{\hat{S}}^{n}}\left[d(S^{n},{\hat{S}}^{n})\right] \end{array}$$
(184)$$\begin{array}{cc} \; & =\frac{1}{n}\sum_{t=1}^{n}E_{S_{t},{\hat{S}}_{t}}\left[d(S_{t},{\hat{S}}_{t})\right]. \end{array}$$
This metric captures the average quality of the reconstructed sequence, making it suitable for applications where consistent strict fidelity for each symbol is not essential but where the overall fidelity of the transmission needs to remain within acceptable limits.

In the case of a deterministic ID code, the average distortion can be expressed in a more detailed form as
(185)$${\bar{d}}^{n}=\frac{1}{N}\sum_{i=1}^{N}E_{S^{n}Y^{n}}\left[\frac{1}{n}\sum_{t=1}^{n}d\left(S_{t},{\hat{S}}_{t}\right)|X^{n}=f_{i}\right].$$
Using the code construction method from Section 3.5 along with the minimum distortion condition defined in (17), we propose the following theorem, which provides a lower bound on the deterministic ID capacity–distortion function for a state-dependent channel $W_{S}$ under an average distortion constraint.

**Theorem 6.** 
*The deterministic ID capacity–distortion function with average distortion constraint ${\bar{d}}^{n}\leq D$ of the state-dependent channel $W_{S}$ is lower-bounded as follows:*

(186)
$$C_{ID,avg}^{d}\left(D\right)\geq \max_{x\in X_{D}}H\left(E[W_{S}\left(\cdot |x,S\right)]\right),$$

*where the set $X_{D}$ is provided by*

(187)
$$X_{D}=\left\{x\in X,\;d^{★}\left(x\right)\leq D\right\}.$$



Despite the practical implications of this result, proving a converse theorem for this bound remains an open problem.

## 5. Conclusions and Discussion

In this work, we have studied the problem of joint ID and channel state estimation over a DMC with i.i.d. state sequences where the sender simultaneously sends an identity and senses the state via a strictly causal channel output. After establishing the capacity on the corresponding ID capacity–distortion function, it emerges that sensing can increase the ID capacity. In the proof of our theorem, we noticed that the generation of common randomness is a key tool for achieving a high ID rate. The common randomness generation is helped by feedback. The ID rate can be further increased by adding local randomness at the sender.

Our framework closely mirrors the one described in [23], with the key distinction being that we utilize an identification scheme instead of the classical transmission scheme. We want to simultaneously identify the sent message and estimate the channel’s state. As noted in the results of [23], the capacity–distortion function is consistently smaller than the transmission capacity of the state-dependent DMC except when the distortion is infinite. This observation aligns with expectations for the message transmission scheme, as the optimization is performed over a constrained input set defined by the distortion function. However, this does not directly apply to the ID scheme. An interesting aspect is that the capacity–distortion function for the deterministic encoding case scales double exponentially with the blocklength, as highlighted in Theorem 4. However, the ID capacity of the state-dependent DMC with deterministic encoding scales only exponentially with the blocklength. This is because feedback significantly enhances the ID capacity, enabling double-exponential growth of the ID capacity for the state-dependent DMC, as established in Theorem 2. This contrasts sharply with the message transmission scheme, where feedback does not increase the capacity of a DMC. Introducing an estimator into our framework naturally reduces the ID capacity compared to the scenario with feedback but without an estimator. This reduction occurs because the optimization is performed over a constrained input set defined by the distortion function. Nevertheless, the capacity–distortion function remains higher than in the case without feedback and without an estimator. This difference underscores a unique characteristic of the ID scheme, highlighting its distinct scaling behavior and potential advantages in certain scenarios.

We consider two cases, namely, deterministic and randomized identification. For a transmission system without sensing, it was shown in [1,3] that the number of messages grows exponentially, i.e., $N=2^{nC_{ID}^{d}}$.

Remarkably, Theorem 4 demonstrates that by incorporating sensing, the growth rate of the number of messages becomes double exponential ($N=2^{2^{nC_{ID}^{d}\left(D\right)}}$); this result is notable, and closely parallels the findings on identification with feedback in [14].

In the case of randomized identification, Theorem 5 shows that the capacity is also improved by incorporating sensing. However, in both the deterministic and randomized settings, the scaling remains double exponential.

One application of message identification is in control and alarm systems [10,33]. For instance, it has been shown that identification codes can be used for status monitoring in digital twins [34]. In this context, our results demonstrate that incorporating a sensing process can significantly enhance the capacity.

Another potential application of our framework is molecular communication, where nanomachines use identification codes to determine when to perform specific actions such as drug delivery [35]. In this context, sensing the position of the nanomachines can enhance the communication rate. For such scenarios, it is also essential to explore alternative channel models such as the Poisson channel.

Furthermore, it is clear that in other applications it would be necessary to consider different distortion functions.

In the future, it would be interesting to apply the method proposed in this paper to other distortion functions. Furthermore, in practical scenarios, there are models where the the sensing is performed either additionally or exclusively by the receiver. This suggests the need to study a wider variety of system models. For wireless communications, the Gaussian channel is more practical and widely applicable. Therefore, it would be valuable to extend our results to the Gaussian case (JIDAS scheme with a Gaussian channel as the forward channel). It has been shown in [15,28] that the ID capacity of a Gaussian channel with noiseless feedback is infinite. Interestingly, the ID capacity of a Gaussian channel with noiseless feedback remains infinite regardless of the scaling used for the rate, e.g., double exponential, triple exponential, etc. By introducing an estimator, we conjecture that the same results will hold, leading to an infinite capacity–distortion function. Thus, considering scenarios with noisy feedback is more practical for future research.

## Figures and Tables

**Figure 1 entropy-27-00012-f001:**
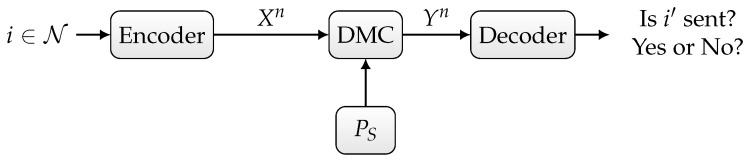
Discrete memoryless channel with random state.

**Figure 2 entropy-27-00012-f002:**
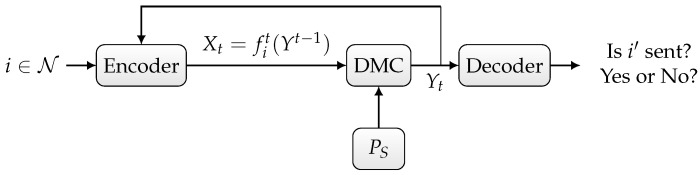
Discrete memoryless channel with random state and noiseless feedback.

**Figure 3 entropy-27-00012-f003:**
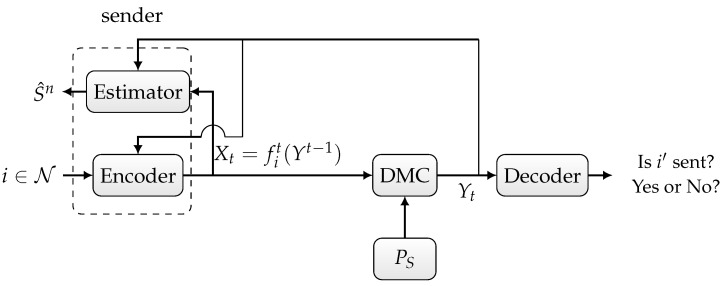
State-dependent channel with noiseless feedback.

## Data Availability

The original contributions presented in this study are included in the article. Further inquiries can be directed to the corresponding author.

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
