# Peer review of "Joint Identification and Sensing for Discrete Memoryless Channels"

_entropy, 2024, doi:10.3390/e27010012_

Round 1

Reviewer 1 Report

Comments and Suggestions for Authors

The authors have investigated joint identification and sensing for discrete memoryless channels (DMCs). This topic is both interesting and timely. Additionally, the authors have provided a complete characterization of the ID capacity-distortion function. Overall, this paper is well-written and demonstrates a high degree of technical rigor. However, there are two minor points that should be addressed to enhance the work:

1: It is recommended to include some discussion or preliminary thoughts on extending the DMC framework to Gaussian channels.
2: The proof of Proposition 5 is not clearly presented. Both the direct and converse proofs should be reorganized for better clarity and comprehension

Author Response

Reviewer 1
(1) It is recommended to include some discussion or preliminary thoughts on extending the
DMC framework to Gaussian channels.
i
Thank you for your comment. In the revised version, we included some discussion on the extension in
the conclusion.
(2) The proof of Proposition 5 is not clearly presented. Both the direct and converse
proofs should be reorganized for better clarity and comprehension.
We appreciate the reviewer’s suggestion. In the revised version, we provided a detailed proof of proposition
5.

Reviewer 2 Report

Comments and Suggestions for Authors

The authors have conducted research on joint identification and sensing over DMCs. Some novel results are provided that are mathematically proved. Overall this paper is well-written and the considered topic is well-motivated from the practical needs for next-generation wireless applications. Furthermore, the reviewers did not find technical flaws from the derivations and believe that the paper is quite ready to publish from its current form. An addition comment is to provide some high-level discussions comparing the proposed scheme with the results of [28], highlighting the potential similarities and differences between the ID codes and the conventional Shannon-type of codes in the context of joint communications and sensing.

Author Response

Reviewer 2
The authors have conducted research on joint identification and sensing over DMCs. Some novel results are
provided that are mathematically proved. Overall this paper is well-written and the considered topic is wellmotivated
from the practical needs for next-generation wireless applications. Furthermore, the reviewers did
not find technical flaws from the derivations and believe that the paper is quite ready to publish from its current
form.
(1) An addition comment is to provide some high-level discussions comparing the proposed scheme with
the results of [28], highlighting the potential similarities and differences between the ID codes and the
conventional Shannon-type of codes in the context of joint communications and sensing. [28] Kobayashi,
M.; Caire, G.; Kramer, G. Joint State Sensing and Communication: Optimal Tradeoff for a Memoryless
Case. In 549 Proceedings of the 2018 IEEE International Symposium on Information Theory (ISIT),
2018, pp. 111–115. https://doi.org/10.110 550 9/ISIT.2018.8437621. 551
Thank you for the comment. In the revised version, we included a comprehensive comparison between
our work on joint identification and sensing and the results presented in [28]. We provided a more
detailed analysis of the similarities between the two frameworks, highlighting their shared principles
and methodologies. Additionally, we elaborated on the potential advantages of our joint identification
and sensing scheme over the approach proposed in [28], emphasizing its ability to deliver enhanced
performance and efficiency under specific scenarios.
